# Mesoporous Pt@Pt-skin Pt₃Ni core-shell framework nanowire electrocatalyst for efficient oxygen reduction

Hui Jin[1,4], Zhewei Xu[1,4], Zhi-Yi Hu ⬡[1,4], Zhiwen Yin ⬡[1], Zhao Wang[1], Zhao Deng[1], Ping Wei[1], Shihao Feng[1], Shunhong Dong[1], Jinfeng Liu[1], Sicheng Luo[1], Zhaodong Qiu[1], Liang Zhou ⬡[1], Liqiang Mai ⬡[1], Bao-Lian Su[1,2], Dongyuan Zhao ⬡[3] & Yong Liu ⬡[1] ✉

The design of Pt-based nanoarchitectures with controllable compositions and morphologies is necessary to enhance their electrocatalytic activity. Herein, we report a rational design and synthesis of anisotropic mesoporous Pt@Pt-skin Pt₃Ni core-shell framework nanowires for high-efficient electrocatalysis. The catalyst has a uniform core-shell structure with an ultrathin atomic-jagged Pt nanowire core and a mesoporous Pt-skin Pt₃Ni framework shell, possessing high electrocatalytic activity, stability and Pt utilisation efficiency. For the oxygen reduction reaction, the anisotropic mesoporous Pt@Pt-skin Pt₃Ni core-shell framework nanowires demonstrated exceptional mass and specific activities of 6.69 A/mg$_{pt}$ and 8.42 mA/cm² (at 0.9 V versus reversible hydrogen electrode), and the catalyst exhibited high stability with negligible activity decay after 50,000 cycles. The mesoporous Pt@Pt-skin Pt₃Ni core-shell framework nanowire configuration combines the advantages of three-dimensional open mesopore molecular accessibility and compressive Pt-skin surface strains, which results in more catalytically active sites and weakened chemisorption of oxygenated species, thus boosting its catalytic activity and stability towards electrocatalysis.

The development of noble metal nanocrystals (NCs) with controlled sizes, compositions, and nanostructures has opened enormous possibilities for the engineering of catalysts with superior activity and selectivity[1–4]. Pt-based NCs with well-defined nanostructures and compositions have been demonstrated to be the most active electrocatalysts for the oxygen reduction reaction (ORR) in fuel cells and metal-air batteries[5–8]. Carpenter et al. have reported the preparation of well-faceted Pt alloy NCs (including cubic and octahedral Pt-Ni alloy NCs) with high ORR activity[9]. However, these solid Pt alloy NCs

contain a substantial proportion of noble metals (Pt) in the bulk than at the surface, which limits the noble-metal utilisation and their commercial applications[10–12]. To date, extensive efforts have been devoted to designing three-dimensional (3D) open Pt and/or Pt-based alloy nanostructures, including hollow and porous nanoparticles such as nanocages and nanoframes, which help reduce the Pt content by maximising the activity and atomic utilisation by exposing both interior and exterior surface[13–16]. However, such 3D open Pt and/or Pt-based alloy nanostructures still suffer from

[1]International School of Materials Science and Engineering (ISMSE), Nanostructure Research Centre, State Key Laboratory of Advanced Technology for Materials Synthesis and Processing, Wuhan University of Technology, Wuhan 430070, China. [2]Laboratory of Inorganic Materials Chemistry, Department of Chemistry, University of Namur, 61 rue de Bruxelles, B-5000 Namur, Belgium. [3]Department of Chemistry, Laboratory of Advanced Materials, Shanghai Key Lab of Molecular Catalysis and Innovative Materials, State Key Laboratory of Molecular Engineering of Polymers, Fudan University, Shanghai 200433, PR China. [4]These authors contributed equally: Hui Jin, Zhewei Xu, Zhi-Yi Hu ✉e-mail: liuyong3873@whut.edu.cn

insufficient catalytic durability owing to quick structural collapse or transformation under detrimental high-temperature or corrosive catalytic conditions[17,18].

Noble metal nanostructures possessing one-dimensional (1D) anisotropic morphologies may solve this inherent stability problem[19,20]. The anisotropic nature of 1D nanostructures can facilitate a close surface contact with the carbon support. Such close contact enhances the electron transfer between the reactants and the Pt surface, and facilitates the binding between the 1D nanostructures (e.g., nanowires) and the carbon support, thus resulting in both high activity and stability[21-23]. Luo et al. have reported 1D mesoporous Pd@Ru core-shell nanorods, which exhibit the most competitive hydrogen evolution reaction (HER) catalytic activity and stability[24]. However, 1D Pt nanostructures still face challenges in optimising the Pt utilisation efficiency and specific activity by engineering their surface structure[25]. Particularly, the surface lattice strain (compressive and/or tensile) of the Pt-skin surface can alter the surface electronic structure and weak chemisorption of oxygenated species[26-28], which is considered an effective approach to enhance ORR activity.

Therefore, the integration of a 3D open porous configuration, 1D anisotropic motif, and lattice-strained Pt-skin surface into one Pt-based nanostructure can be predicted to be beneficial for the development of long-term active electrocatalysts with maximised Pt utilisation efficiency. Herein, we demonstrate this concept by synthesising well-defined anisotropic mesoporous Pt@Pt-skin Pt$_3$Ni core-shell framework nanowires (CSFWs). The mesoporous Pt@Pt-skin Pt$_3$Ni CSFWs configuration was rationally designed to combine the advantages of a 1D ultrathin atomic-jagged Pt nanowire (~3 nm in diameter) core and a 3D open lattice-strained Pt-skin Pt$_3$Ni framework shell, endowing high activity, stability, and Pt utilisation efficiency. In particular, the presence of mesopores in the 3D open lattice-strained Pt-skin Pt$_3$Ni framework shell facilitated highly exposed surface areas, which maximised the use of Pt atoms and sped up the reactant transport. As

expected, the mesoporous Pt@Pt-skin Pt$_3$Ni CSFWs catalyst exhibited superior electrocatalytic performance for the technologically important ORR in hydrogen fuel cells. Moreover, it exhibited exceptional mass and specific activities of 6.69 A/mg$_{pt}$ and 8.42 mA/cm$^2$ (at 0.9 V versus reversible hydrogen electrode). The catalyst also exhibited superior stability with negligible activity decay (less than 3%) after 50,000 cycles.

## Results

### Material synthesis and structural characterizations

Figure 1a schematically illustrates the synthetic strategy for anisotropic mesoporous Pt@Pt-skin Pt$_3$Ni CSFWs. We demonstrated the configuration of CSFWs by the solvothermal reduction of platinum (II) acetylacetonate [Pt(acac)$_2$] and nickel (II) acetylacetonate [Ni(acac)$_2$] in an oleylamine/octadecene/glucose/cetyltrimethylammonium bromide (CTAB) mixture at 200 °C (oleylamine/octadecene mixture acted as solvents/surfactants, glucose as the reducing agent and CTAB as the capping agent; see details in the supplementary materials). As the reduction potential of Pt$^{2+}$/Pt is more positive than Ni$^{2+}$/Ni under the same reaction conditions (see Supplementary Fig. 1 for a comparison of the reduction ability), the Pt precursor first gradually reduces to yield ultrathin pure jagged Pt nanowires (NWs) with abundant surface atomic steps (Fig. 1b). Such intrinsic surface atomic steps allow further site-selective nucleation of the Pt-Ni alloy phase on Pt nanowires (Fig. 1c) and induce the formation of uniform nanogourd-string-like Pt@Pt-Ni alloy core-shell nanowires (CSNWs) (Fig. 1d). After treatment under acidic conditions (acetic acid), a Ni-rich phase within the nanogourd-string-like Pt-Ni alloy shell is selectively etched, forming well-defined anisotropic mesoporous Pt@Pt$_3$Ni CSFWs. The mesoporous Pt@Pt$_3$Ni CSFWs are then annealed in an argon/hydrogen mixture (Ar/H$_2$: 97:3) at 300°C to simultaneously form Pt-skin surface and remove organic surfactants[29], yielding well-defined mesoporous Pt@Pt-skin Pt$_3$Ni CSFWs (Fig. 1e). The overall morphology of the

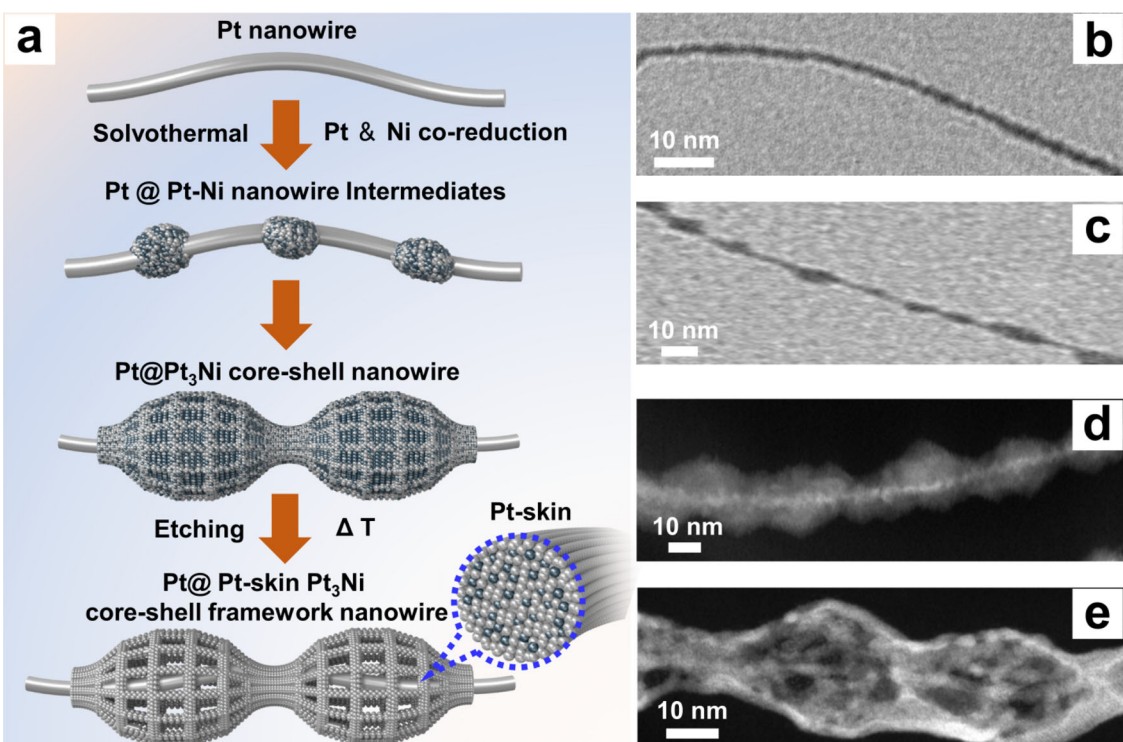

**Fig. 1 | Illustration (left) and corresponding TEM images (right) of the samples obtained at four stages during the growth process from ultrathin atomic-jagged Pt nanowires to mesoporous Pt@Pt-skin Pt$_3$Ni core-shell framework nanowires (CSFWs). a** Schematic illustration of the preparatiom of mesoporous Pt@Pt$_3$Ni CSFWs, where ΔT represents the annealing treatment. **b** TEM image of a single ultrathin atomic-jagged Pt nanowire. **c** TEM image of a single Pt@PtNi nanowire intermediate. **d** TEM image of a single Pt@Pt-Ni alloy CSNWs. **e** TEM image of final mesoporous Pt@Pt-skin Pt$_3$Ni CSFWs.

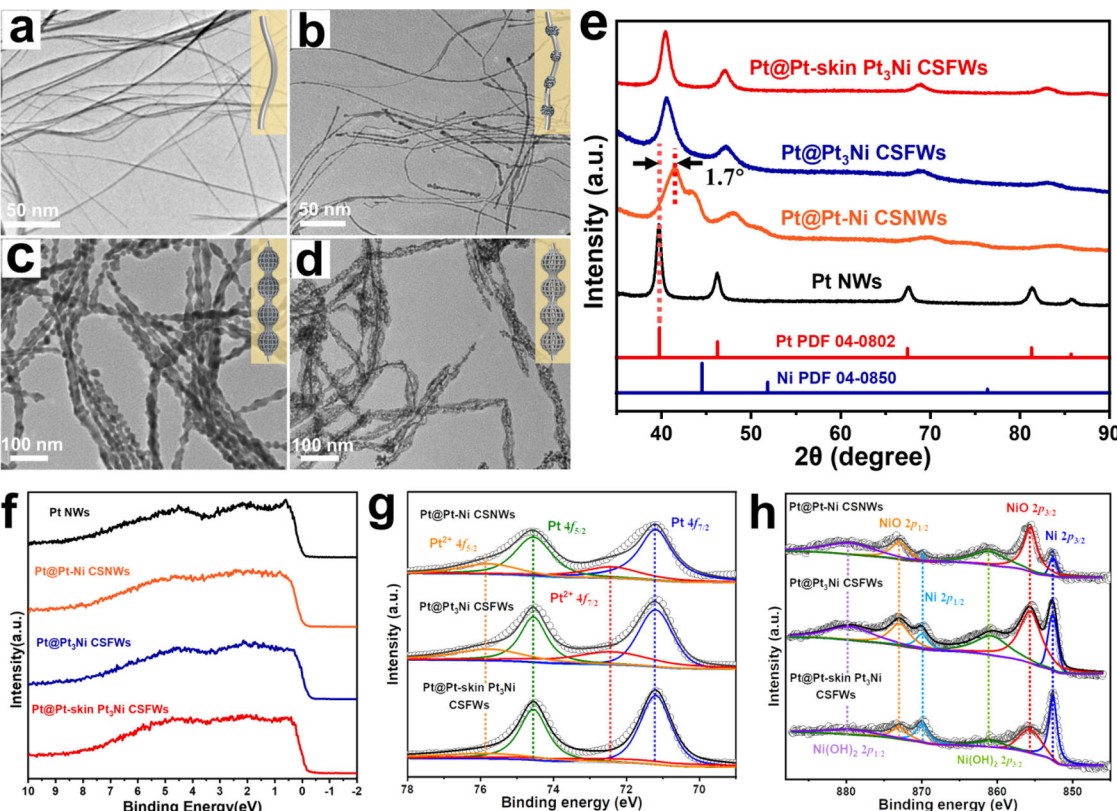

**Fig. 2 | Morphological and structural characterizations of various Pt-based samples collected from the reactions at different stages.** Typical TEM images of **a** initial ultrafine Pt nanowire. **b** Pt@PtNi nanowire intermediates. **c** Pt@Pt-Ni alloy CSNWs. **d** Final mesoporous Pt@Pt-skin Pt$_3$Ni CSFWs. **e** Ex-situ XRD patterns of the samples evolved from Pt nanowires into mesoporous Pt@Pt-skin Pt$_3$Ni CSFWs. **f** High-resolution valence band UPS of all samples. **g, h** Pt 4$f$ and Ni 2$p$ XPS patterns of Pt@PtNi CSNWs, Pt@Pt$_3$Ni CSFWs and mesoporous Pt@Pt-skin Pt$_3$Ni CSFWs.

CSFWs was maintained without obvious changes in length or diameter during the acid etching and annealing treatments (Fig. 1e).

We tracked the structural evolution and intermediate structure composition during solution phase growth. The transmission electron microscopy (TEM) images showed that the initially formed Pt nanowires (collected after a growth period of 30 min) had a typical overall diameter of ~3 nm (Fig. 2a and Supplementary Fig. 2a, b). High-resolution TEM (HRTEM) images further revealed that the Pt nanowires had a typical Pt (111) lattice spacing of 0.23 nm and rough surface with a high density of surface atomic steps (Supplementary Fig. 2c)[30]. Energy-dispersive-X-ray (EDX) spectroscopy elemental analysis confirmed that the Pt was the only element in the initial resulting ultrathin nanowires (Supplementary Figs. 3 and 4). With the gradual depletion of Pt ions and catalytic effect of the initially reduced Pt on Ni reduction[31,32] (Supplementary Fig. 1), Pt deposition was considerably suppressed, and the Pt-Ni alloy phase began to selectively deposit on the defective surface sites of the Pt nanowires, as evidenced by the TEM/HRTEM images (Fig. 2b and Supplementary Fig. 5). When the synthetic reaction was extended to 24 h, the TEM images showed that the overall morphology changed to well-defined nanogourd-string-like Pt@Pt-Ni alloy CSNWs (Fig. 2c and Supplementary Fig. 6). According to the high-angle annular dark-field scanning transmission electron microscopy (HAADF-STEM) and TEM-EDS analyses, the Pt@Pt-Ni alloy CSNWs exhibited an apparent core/shell structure with an overall Pt/Ni atomic ratio of approximately 1:4.3 (Supplementary Fig. 7). HRTEM image (Supplementary Fig. 8) of Pt@Pt-Ni alloy CSNWs showed lattice fringes of 0.22 nm at the shell-edges (or corners) and 0.21 nm at the shell surface corresponding to (111) planes of Pt-rich phase and Ni-rich phase, indicating Pt-rich edges and Ni-rich surfaces in nanogourd-like Pt-Ni alloy shell[33]. The above results suggest that the large Pt atoms

(1.39 Å for Pt, 1.24 Å for Ni) on the interior of Pt-Ni alloy shell preferentially migrate outwards to the vertices and/or edge sites to relieve the significant lattice strain energy under solvothermal conditions[34]. After the acetic acid treatment, the Ni-rich phase within the nanogourd-like Pt-Ni alloy shell was selectively etched, resulting in a well-defined 3D open mesoporous Pt$_3$Ni framework shell anchored on the ultrathin Pt nanowire (Fig. 2d). Controlled thermal treatment of the resulting Pt-rich shell formed Pt-skin surface nanostructure——mesoporous Pt@Pt-skin Pt$_3$Ni CSFWs (Supplementary Fig. 9), which was likely guided by the energetic favourability of larger Pt atoms migrating from the interior of the Pt-rich phase to the surface of the Pt$_3$Ni framework shell[35]. As shown in Supplementary Fig. 10, the average diameters of the initially formed Pt nanowires, Pt@Pt-Ni alloy CSNWs, and final mesoporous Pt@Pt-skin Pt$_3$Ni CSFWs is calculated to be 3.3 ± 1 nm, 24.3 ± 3 nm and 20.3 ± 3 nm, respectively.

We further studied the effects of the temperature, reaction time, concentration of the various reagents (including CTAB and glucose) and the Pt/Ni molar ratio in the synthesis solution on the formation of the structures[36] (Supplementary Figs. 11 to 15). As shown in Supplementary Fig. 12, the temperature-dependent morphology changes results revealed that the length of the nanogourd-string-like Pt@Pt-Ni alloy CSNWs increases from tens of nanometres (160 °C) to hundreds of nanometres (180 °C), and finally increases to a few microns (200 °C). However, when the reaction temperature further increases to 220 °C, an uneven nanogourd-string-like Pt@Pt-Ni alloy CSNWs and nanoparticles mixture is formed. The use of CTAB also plays a critical role in determining the morphology of the Pt-Ni nanowires. As shown in Supplementary Fig. 13, without the addition of CTAB, only irregular polyhedral nanocrystals (NCs) with an average size of 30 ± 5 nm are produced. After the addition of 10 mg of CTAB, some nanogourd-

string-like Pt@Pt-Ni CSNWs mixed with NCs are observed. Uniform nanogourd-string-like Pt@Pt-Ni CSNWs (with average diameters of 24.3 nm) are obtained in yields with 40 mg of CTAB. When the CTAB amount is further increased to 50 mg, the yield of Pt@Pt-Ni CSNWs decreases, and the morphology becomes irregular. Similarly, the use of glucose as a reductant is also important for the formation of well-defined Pt@Pt-Ni CSNWs (Supplementary Fig. 14). We further studied the effect of Ni/Pt molar ratio on the formation of the structures. As shown in Supplementary Fig. 15, only Pt nanowires with smooth surfaces are obtained in the absence of Ni(acac)$_2$. When the Ni/Pt molar ratio is increased to 0.46 by adding 3mg Ni(acac)$_2$, the Pt nanowires are incorporated into the thin Pt@Pt-Ni nanowires. Typical uniform nanogourd-string-like Pt@Pt-Ni CSNWs (with average diameters of 24.3 nm) are obtained with a Ni/Pt molar ratio of approximately 1.24 [by adding 8 mg Ni(acac)$_2$]. Upon further increasing the Ni/Pt molar ratio to 2.14, uneven and agglomerated Pt@Pt-Ni CSNWs with diameters of 50 ± 8 nm are obtained. Thus, the morphology studies of the catalyst confirm that the optimum reaction temperature, time, concentrations of reagents (CTAB, glucose) and Ni/Pt molar ratio are responsible for the fine-controlled production of the Pt@Pt-Ni CSFWs.

The phase evolution from Pt nanowires into Pt@Pt-Ni alloy CSNWs and eventually into mesoporous Pt@Pt-skin Pt$_3$Ni CSFWs was further studied by ex-situ X-ray diffraction (XRD). As shown in Fig. 2e, the XRD diffraction peaks of pure Pt nanowires are characteristic of typical face-centered cubic (fcc), which is consistent with the HRTEM results. After selective nucleation of the Pt-Ni alloy phase on the defective surface of the Pt nanowires, the corresponding diffraction peaks of the Pt@Pt-Ni alloy CSNWs shift to higher 2θ angles (approximately 1.7° positive shift) owing to the decreased lattice constant upon the addition of Ni (Fig. 2e and Supplementary Fig. 16)[37]. The asymmetric peaks of Pt@Pt-Ni alloy CSNWs can be split into two sets of diffraction patterns, which are assigned to the Pt-rich phase (Pt$_3$Ni) and a Ni-rich phase (PtNi$_4$), respectively. However, the asymmetric peak becomes a single set of symmetric peaks, and the peaks shift back to a lower angle (approximately 1.0° negative-shift) after acid etching, which suggests that the composition has changed from the alloyed phase to a single Pt$_3$Ni phase. The final thermal treatment in the argon/hydrogen mixture at 300°C is induces a transition from the Pt$_3$Ni-skeleton into the Pt-skin structure by surface segregation and restructuring[29,38]. When compared with pure Pt nanowires, the XRD diffraction peaks of Pt@Pt-skin Pt$_3$Ni CSFWs and Pt@Pt$_3$Ni CSFWs have shifted to higher angles with wider widths, which could be attributed to the decrease in lattice distance when smaller Ni atoms alloyed with Pt atoms in the lattice, resulting in lower crystallinity.

The evolution of the electronic band structure of the samples was further investigated using high-resolution UPS and XPS. The UPS results in Fig. 2f and Supplementary Table 1 showed that the d-band center (the detailed calculations of d-band center are described in Supporting Information) decreases from −2.42 eV (Pt NWs) to −2.49 eV (Pt@Pt-Ni alloy CSNWs) after the Pt-Ni alloying process, which originates from the electronic interaction between Pt and the alloyed Ni atoms. Moreover, the d-band center of Pt@Pt-skin Pt$_3$Ni CSFWs exhibits -0.32 eV downshift compared with Pt nanowires, indicating a Pt-skin lattice compression on the surface of the annealed Pt@Pt-skin Pt$_3$Ni CSFWs (the d-band center of Pt is highly sensitive to lattice strain, shifting positively as the lattice expands and negatively as the lattice contracts). The observed decrease in the d-band center position is attributed to the reduction in the adsorption energies of the oxygenated species on the Pt-skin surface, which can significantly optimise the catalytic performance of the Pt@Pt-skin Pt$_3$Ni CSFWs for electrochemical reactions(e.g. ORR)[39,40]. Furthermore, the Ni 2$p$ and Pt 4$f$ XPS spectra of the Pt@Pt-Ni alloy CSNWs showed that most of the surface Ni was oxidized and the surface Pt was mainly in the metallic state (Fig. 2g–h). After acid etching, the intensities of Pt 4$f$ and Ni 2$p$ dramatically increased, whereas the ratio of Ni$^{x+}$ at the surface

substantially decreased, implying a preferentially etching of low-coordinated Ni at the Ni-rich surface. The precise Pt-Ni atomic ratio of the final mesoporous Pt@Pt-skin Pt$_3$Ni CSFWs was determined to be approximately 3.4:1 by XPS, which is consistent with the ICP-OES and EDS results (Supplementary Fig. 17).

The detailed atomic structure of the mesoporous Pt@Pt-skin Pt$_3$Ni CSFWs was characterised using aberration-corrected HAADF-STEM. The HAADF-STEM image of the individual mesoporous Pt@Pt-skin Pt$_3$Ni CSFWs clearly shows an ultrafine (-3 nm) Pt nanowire core and 3D open mesopores (2–5 nm) are well exposed on the nanogourd-string-like Pt$_3$Ni framework shell (Fig. 3a, b, d and Supplementary Fig. 18a–c). The STEM-energy-dispersive X-ray spectroscopy (EDX) intensity profile (Fig. 3e, g, h and Supplementary Fig. 18d–i) shows that Ni is mainly distributed on the mesoporous hollow framework shells of the Pt@Pt-skin Pt$_3$Ni CSFWs, whereas Pt is mainly distributed in the central Pt nanowire cores and surface of the mesoporous Pt$_3$Ni framework shells. The high-resolution HAADF-STEM and the corresponding fast Fourier transform (FFT) patterns measured along the [$\bar{1}$01] zone axis (Supplementary Fig. 19) show a high-quality periodic lattice of face-centered cubic (fcc) Pt extending across the entire surface of the Pt@Pt-skin Pt$_3$Ni CSFWs, indicating the formation of Pt-skin surface nanostructures. The simulated Pt atomic model along the [$\bar{1}$01] zone axis (with 4.9% compressive lattice contraction) matches considerably well with the experimental atomic arrangement obtained from the selected areas (marked by the rectangles in Supplementary Fig. 19). Atomic layer-by-layer EDX line scans (Fig. 3f, i) clearly showed only a Pt signal in the second outermost atomic layer (but the Pt and Ni mixed signal appears in the tenth outermost atomic layer), further confirming that a well-defined Pt-skin structure was formed on the surface of ultrathin Pt$_3$Ni curved framework shells. The thickness of the Pt-skin was -1–1.5 nm, corresponding to roughly 5–8 atomic Pt layers; such ultrathin Pt-skin on Pt$_3$Ni have considerable potential as an active catalyst with high Pt atom utilization efficiency. Compared with bulk Pt (111) spacing values (2.27 Å), the intensity profile of Pt-skin shows a much smaller Pt (111) lattice distance (2.16 Å) and compressive lattice contraction -4.9% (Fig. 3c), which is consistent with high-resolution HAADF-STEM result. We emphasise that the ultrathin curved hollow framework walls within CSFWs not only endow a high electrochemically active surface area but also induce compressive strain, which can alter their surface electronic band structure and in turn boost their electrochemical activity[41].

## ORR performance evaluation

The anisotropic Pt@Pt-skin Pt$_3$Ni CSFWs configurations would be beneficial for electrocatalysis due to the integration of the 1D atomic-jagged Pt nanowire-core and 3D open mesoporous Pt-skin Pt$_3$Ni hollow framework shells. The electrocatalytic properties of the mesoporous Pt@Pt-skin Pt$_3$Ni CSFWs/C were evaluated and benchmarked against 1D directly synthesised Pt nanowires/C, Pt@Pt$_3$Ni CSFWs/C, and the commercial Pt/C nanoscale electrocatalysts (20 wt% Pt on a Vulcan XC-72 carbon support, Pt particle size of 2–5 nm) (Fig. 4). Before the electrochemical measurements, all the catalysts were uniformly deposited on a commercial carbon (C, Vulcan) support and then loaded onto glassy carbon electrodes. The ORR polarisation curves of the four samples in Fig. 4a exhibit a positive shift in the following order: commercial Pt/C < Pt nanowires/C < Pt@Pt$_3$Ni CSFWs/C < mesoporous Pt@Pt-skin Pt$_3$Ni CSFWs/C. The cyclic voltammetry (CV) curves (inset of Fig. 4a) show two distinctive potential regions associated with the underpotentially deposited hydrogen (H$_{UPD}$, H$^+$ + e$^-$ = H$_{UPD}$, 0 < E < 0.37 eV) and absorbed hydroxyl species (OH$_{ad}$, 2H$_2$O = OH$_{ad}$ + H$_3$O$^+$ + e$^-$, E > 0.7 eV)[42]. Additionally, the CV curves in Fig. 4a show that the onset potential of the formation of OH$_{ad}$ on mesoporous Pt@Pt-skin Pt$_3$Ni CSFWs/C exhibits a distinct positive shift in OH$_{ad}$ compared with pure Pt (Pt nanowires/C and the commercial Pt/C) and Pt@Pt$_3$Ni CSFWs/C, suggesting a relatively weaker chemisorption energy of OH$_{ad}$ on the Pt-

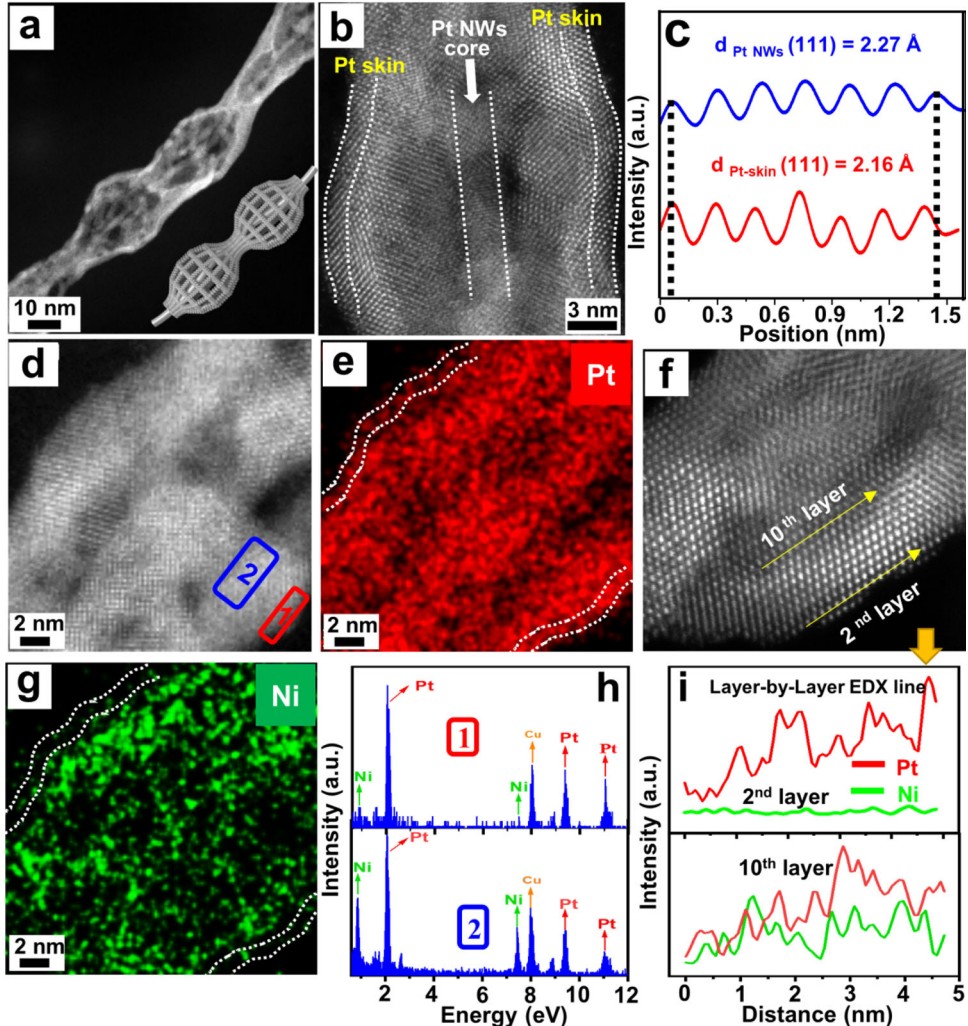

**Fig. 3 | Structural and compositional characterizations of mesoporous Pt@Pt-skin Pt₃Ni CSFWs.** **a**, **b**, **d**, **f** HAADF-STEM and HRTEM images. **c** Intensity profiles of bulk Pt (111) and Pt-skin (111), which represents ~4.9% compressive strain of Pt-skin (111) compared to bulk Pt (111). **e**, **g** STEM-EDX elemental mapping. **h** EDX intensity profile taken from rectangular marked area in **d**. **i** Layer-by-layer EDX line scan on outmost framework shells marked in **f**. The above EDX results clearly revealed a well-defined Pt-skin surface was formed on the surface of ultrathin mesoporous Pt₃Ni curved framework shells.

skin surface of the Pt@Pt-skin Pt₃Ni CSFWs/C[43]. Subsequent Koutecky-Levich (K-L) plots of mesoporous Pt@Pt-skin Pt₃Ni CSFWs/C originating from polarisation curves at different rotating rates (Supplementary Fig. 20) present a good linearity relationship between $J^{-1}$ and $\omega^{-1/2}$, indicating the first-order reaction kinetics for ORR towards the concentration of dissolved $O_2$. The number of transfer electrons calculated from the K-L equation was approximately 4.07 for the mesoporous Pt@Pt-skin Pt₃Ni CSFWs/C, which suggested an efficient 4-electrons reaction process on the mesoporous Pt@Pt-skin Pt₃Ni CSFWs/C catalyst.

The electrochemical active surface area (ECSA) calculated by $H_{UPD}$ of mesoporous Pt@Pt-skin Pt₃Ni CSFWs/C is 79.45 m²/g$_{Pt}$, which is substantially higher than that of Pt nanowires/C (62.34 m²/g$_{Pt}$), Pt@Pt₃Ni CSFWs/C (68.22 m²/g$_{Pt}$) and commercial Pt/C (70.04 m²/g$_{Pt}$), demonstrating the enhanced ECSA of mesoporous Pt@Pt-skin Pt₃Ni CSFWs/C with the integration of 1D atomic-jagged nanowire-core and 3D open mesoporous Pt-skin hollow-shell configuration. As shown in Fig. 4b, the Tafel plots of specific activity exhibit slopes of 51.9, 63.4, 69.2 and 80.9 mV dec⁻¹ for Pt@Pt-skin Pt₃Ni CSFWs, Pt@Pt₃Ni CSFWs, Pt NWs and Pt/C electrocatalyst, respectively. The considerably smaller Tafel slope achieved in the Pt@Pt-skin Pt₃Ni CSFWs suggests significantly improved ORR kinetics. We also measured the ECSA by the electrooxidation of carbon monoxide (CO stripping) (Supplementary

Fig. 21). Because $CO_{ad}$ has a considerably stronger binding interaction with the Pt-skin surface, the ECSA obtained from CO stripping would be higher than the values derived from $H_{UPD}$. As predicted, the ECSA value of mesoporous Pt@Pt-skin Pt₃Ni CSFWs/C obtained from the CO stripping is 116.6 m²/g$_{Pt}$ (Fig. 4c), and the ratio of ECSA$_{CO}$:ECSA$_{HUPD}$ is calculated to be 1.47 (Supplementary Table 2), which strongly confirms the formation of Pt-skin surface on mesoporous Pt@Pt-skin Pt₃Ni CSFWs/C[44,45]. As shown in Fig. 4d, the mesoporous Pt@Pt-skin Pt₃Ni CSFWs/C delivers a high specific activity (SA) of 8.42 mA cm⁻² at 0.9 V versus RHE, which is approximately 4.4, 2.6 and 26 times as those of Pt nanowires/C (1.92 mA cm⁻²), Pt@Pt₃Ni CSFWs/C (3.2 mA cm⁻²) and commercial Pt/C (0.33 mA cm⁻²), respectively. The mass activity (MA) of Pt@Pt-skin Pt₃Ni CSFWs/C is 6.69 A mg$_{pt}$⁻¹ at 0.9 V versus RHE, which is 15 times that of the 2020 U.S. Department of Energy (DOE) target (0.44 A mg$_{pt}$⁻¹) at 0.9 V for MEA, which placed it among the most efficient Pt-Ni based bimetallic catalysts recently reported for ORR.

Supplementary Figs. 22–24 and Supplementary Table 3 show the reproducibility of the CV curves, ORR polarizsation curves and specific and mass activity for four samples; each sample is measured for five independent thin-film electrodes in 0.1 M HClO₄ electrolyte. The average mass activity for Pt/C, Pt NWs, mesoporous Pt@Pt₃Ni CSFWs and mesoporous Pt@Pt-skin Pt₃Ni CSFWs is 0.23 ± 0.012, 1.2 ± 0.045, 2.18 ± 0.062 and 6.69 ± 0.083 A mg⁻¹$_{Pt}$, respectively. The average

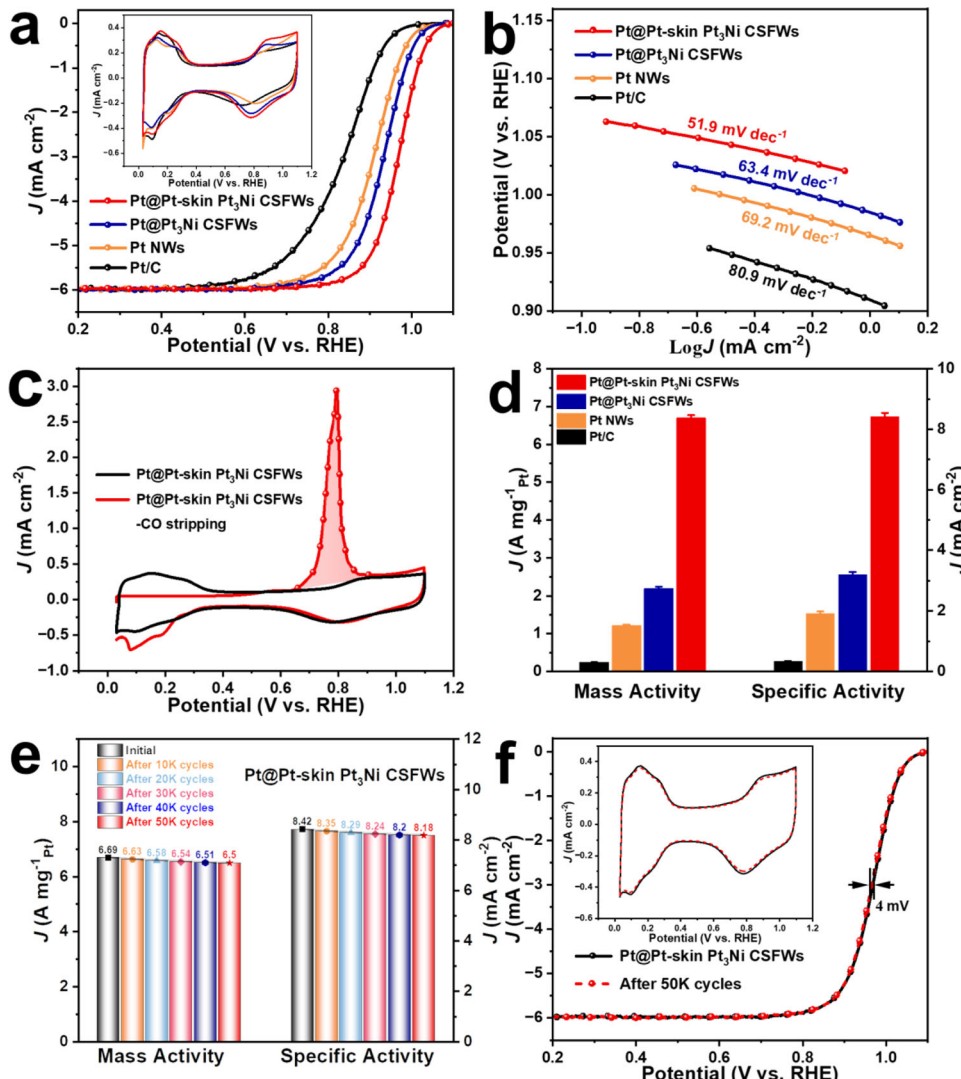

**Fig. 4 | Electrocatalytic performance of mesoporous Pt@Pt-skin Pt₃Ni CSFWs/C, Pt@Pt₃Ni CSFWs/C, Pt NWs/C, commercial Pt/C catalysts for ORR. a** ORR polarization curves. The inset is the CVs of various Pt-based catalysts in 0.1 M HClO₄ solution at a sweep rate of 50 mV/s. **b** Corresponding Tafel plots. **c** CV and CO stripping curves of mesoporous Pt@Pt-skin Pt₃Ni CSFWs/C. **d** Mass and specific activities at 0.9 V (versus RHE). The error bars in **d** are calculated based on five independent test results. **e** Mass and specific activity evolutions for the mesoporous Pt@Pt-skin Pt₃Ni CSFWs/C catalyst before and after different potential cycles. **f** ORR polarization curve evolutions for the mesoporous Pt@Pt-skin Pt₃Ni CSFWs/C catalyst before and after 50,000 potential cycles. The Pt loading for all the catalysts on RDE is 6.5 µg cm⁻².

specific activity for Pt/C, Pt NWs, mesoporous Pt@Pt₃Ni CSFWs and mesoporous Pt@Pt-skin Pt₃Ni CSFWs is 0.33 ± 0.016, 1.92 ± 0.063, 3.2 ± 0.09 and 8.42 ± 0.126 mA cm⁻², respectively. The above rotating disk electrode (RDE) testing clearly confirms that the ORR activity of the catalyst is reproducible and comparable. Moreover, we also evaluated the electrocatalytic performance of different mesoporous Pt@Pt-Ni CSFWs samples obtained with different the Ni/Pt ratios (all catalyst samples underwent the same acid and heat treatments before the ORR test). The ORR results in Supplementary Fig. 25 showed the volcano-shaped ORR activity relationship, and the uniform mesoporous Pt@Pt-skin Pt₃Ni CSFWs/C obtained by adding 8 mg of Ni(acac)₂ achieves the best ORR performance. A comparison of the ORR activity of mesoporous Pt@Pt-skin Pt₃Ni CSFWs/C with other Pt₃Ni/C catalysts published in recent years is also provided in Supplementary Table 4, which shows that the performance of mesoporous Pt@Pt-skin Pt₃Ni CSFWs/C is among the best reported performance for Pt₃Ni/C catalysts.

The electrocatalytic durability of all the catalysts was tested by applying linear potential sweeps between 0.6 and 1.1 V at 100 mV/s in

O₂-saturated 0.1 M HClO₄ solutions. After 50,000 potential-scanning cycles, there was only a 4-mV negative-shift in the half-wave potential for the mesoporous Pt@Pt-skin Pt₃Ni CSFWs/C (Fig. 4f) compared with that of the fresh sample. The SA and MA of the mesoporous Pt@Pt-skin Pt₃Ni CSFW/C decreased by only 2.9 and 2.8% (Fig. 4e), respectively. The TEM images showed negligible changes in the overall morphology and size of the mesoporous Pt@Pt-skin Pt₃Ni CSFWs/C after 50,000 cycles (Supplementary Fig. 30). In addition, as shown in Supplementary Tables 5, 6 and Supplementary Fig. 29, the multilayered Pt-skin surfaces effectively protect the Pt@Pt-skin Pt₃Ni CSFWs/C catalyst against Ni-leaching from the inner region of the framework walls. In contrast, Pt@Pt₃Ni CSFWs/C showed a larger loss in MA (43.6%) and SA (43.8%), along with a −23 mV negative-shifted half-wave potential (Supplementary Fig. 26) after 10,000 cycles. After 10,000 cycles, the Pt nanowires showed an MA loss of 31.7%, an SA loss of 31.3% and a negative-shifted half-wave potential of approximately 16 mV (Supplementary Fig. 27). However, the commercial Pt/C showed a considerably larger negative shift (-73 mV) in the ORR polarisation curves with 47% loss of MA, 48% loss of SA and severe carbon corrosion

(Supplementary Fig. 28) after 10,000 cycles. Commercial Pt/C also exhibited noticeable morphological changes and substantial aggregation after long-term cycling. Compared with Pt@Pt₃Ni CSFWs/C, Pt nanowires/C and commercial Pt/C, we believe that the high catalytic durability of Pt@Pt-skin Pt₃Ni CSFWs originates from their unique structure: (1) the unique electronic structure of the Pt-skin surface may result in a lower coverage of oxygenated intermediates because of the weaker oxygen binding strength, which diminishes the probability of Pt dissolution and the carbon corrosion. (2) The anisotropic porous nature of the Pt@Pt-skin Pt₃Ni CSFWs configuration may lead to multipoint Pt-skin surface contacts with the carbon support. Such close contacts may prevent movement, aggregation and Ostwald ripening processes usually observed in NP_S, and facilitates the binding between the Pt-skin porous CSFWs nanostructures and the carbon support, thereby contributing to the good durability and less carbon corrosion. (3) The optimised Pt-skin thickness of at least four Pt monolayers (MLs) (Fig. 3f and Supplementary Figs. 18 and 19) hinders the loss of subsurface transition metals through the place-exchange mechanism during electrochemical operation, consequently preserving the high intrinsic activity.

## Theoretical calculation and mechanism analysis

To gain further insight into the substantially high ORR activity exhibited by the mesoporous Pt@Pt-skin Pt₃Ni CSFWs, we performed density functional theory (DFT) calculations for the oxygen adsorption energy ($E_O$) on the fcc (111) Pt-skin surface of the mesoporous Pt@Pt-skin Pt₃Ni CSFWs. Typically, a properly weakened Pt-O binding strength leads to superior ORR activity for Pt and its alloys[46]. Thus, we calculated $E_O$ on the fcc Pt (111) surface as a function of strain, which varied from 6% (tensile) to −6% (compressive). Supplementary Fig. 31 shows a volcano-shaped $E_O$-strain relationship with the lowest $E_O$ (−3.6 eV) under −5% compressive lattice strain, which agrees well with our experimentally obtained best ORR performance on the 4.9% compressive (111) Pt-skin surface of mesoporous Pt@Pt-skin Pt₃Ni CSFWs. In addition, the highly atomic-jagged surface of ultrathin Pt nanowire cores (~3 nm) is has been proven to induce a stressed Pt-Pt bond (~2 compressive strain), which can markedly decrease the reaction barrier of the rate-determining steps of the ORR, thus improving specific ORR activity[25]. Furthermore, previous DFT calculations also revealed that the binding energy of the compressive Pt surface atoms on Pt₃Ni is higher than that of pure Pt nanowires[44], which may also be responsible for the higher stability of the mesoporous Pt@Pt-skin Pt₃Ni CSFWs.

## Methods

### Chemicals

Platinum (II) acetylacetonate (Pt(acac)₂, 97%), nickel (II) acetylacetonate (Ni(acac)₂, 95%), cetyltrimethylammonium bromide (CTAB, 99%), glucose (C₆H₁₂O₆, ultra-pure, ≥99.5%), oleylamine (OAm, 80–90%), 1-Octadecene (ODE, GC, > 90.0%) and perchloric acid (HClO₄, AR, 70.0–72.0%) were all purchased from Shanghai Aladdin Chemical Reagent Company. Toluene (C₇H₈, AR, ≥99.5%), acetic acid (C₂H₄O₂, AR, ≥99.5%), ethanol (C₂H₆O, AR, ≥99.7%) and cyclohexane (C₆H₁₂, AR, ≥99.5%) were all purchased from Sinopharm Chemical Reagent Co. Nafion (5%) was purchased from Macklin. Commercial Pt/C catalyst (20 wt %, 2–5 nm Pt nanoparticles) was obtained from Johnson Matthey (JM) Corporation. All the chemicals were used as received without further purification. Deionized water with a resistivity of 18.2 MΩ·cm at 25 °C was used in all experiments.

### Synthesis of Pt@Pt₃Ni and Pt@Pt-skin Pt₃Ni CSFWs

In a typical synthesis of Pt@Pt₃Ni CSFWs, platinum (II) acetylacetonate (Pt(acac)₂, 10 mg), nickel (II) acetylacetonate (Ni(acac)₂, 8 mg), cetyltrimethylammonium bromide (CTAB, 40 mg), glucose (60 mg),

oleylamine (OAm, 5 mL) and 1-Octadecene (ODE, 5 mL) were added into a 25 mL vial and ultrasonicated for 1 h to obtain a homogeneous solution. Subsequently, the solution was transferred into a 25 mL Teflon-lined stainless steel autoclave, which was further heated from 25 °C to 200 °C and kept at 200 °C for 24 h, before it was cooled to room temperature. Finally, the products were collected by centrifugation, using a mixture of ethanol and cyclohexane (volume ratio of 3:1) for three times at 10,278 × g for 10 minutes and the obtained samples were dried at room temperature. To create the anisotropic mesoporous structure, the obtained products were etched in acetic acid (C₂H₄O₂, 2 mL), toluene (C₇H₈, 2 mL) and oleylamine (OAm, 10 uL) mixture solution at 90 °C for 2 h to remove the reactive Ni. After the etching process, the products were collected by centrifugation and washed three times with ethanol and cyclohexane (volume ratio of 3:1), and then dried under ambient conditions. To further obtain the Pt-skin structure, the synthesized Pt@Pt₃Ni CSFWs products were annealed in a tube furnace at 300 °C for 3 hours under the protection of argon-hydrogen mixture gas (volume ratio of 97:3). The resulting samples were cooled to room temperature and collected for subsequent tests.

### Characterization

X-ray diffraction (XRD) characterization was carried out on a Bruker D8 X-ray diffractometer operated at 40 kV and 40 mA, using a Cu-Kα radiation source (λ = 1.54056 Å). Transmission electron microscopy (TEM), energy-dispersive X-ray spectroscopy (EDX), High-angle annular dark-field scanning transmission electron microscopy (HAADF-STEM), and high-resolution transmission electron microscopy (HRTEM) images were obtained with a JEM- 2100F microscope. The X-ray photoelectron spectroscopy (XPS) of the samples was carried out using a Thermo Scientific Kα XPS spectrometer equipped with a monochromatic Al-Kα x-ray source (hv = 1486.6 eV). Ultraviolet photoelectron spectroscopy (UPS) was carried out the same chamber with a He I UV source at a bias voltage of −5V. The d-band center is calculated according to the following equation:[38,47]

$$d \text{ band center} = \frac{\int_{-\infty}^{E_f} R(E)EdE}{\int_{-\infty}^{E_f} R(E)dE} \qquad (1)$$

where E is the binding energy, R(E) is the UPS intensity after background subtraction, $E_f$ is the Fermi energy level, and the calibration of the UPS revealed that the $E_f$ of all samples was approximately 0. The actual Pt loadings in the catalyst were determined by the inductively coupled plasma-optical emission spectrometer (ICP-OES).

### Electrochemical measurement

Before the electrochemical tests, the prepared catalysts were first loaded on the commercial carbon support (XC-72R) to obtain a good dispersion and the loading amount was controlled at 20 wt%. In short, the obtained products and carbon were mixed in cyclohexane and stirred vigorously for 12 h, followed by drying in an oven at 60 °C overnight. Then, 1.27 mg of carbon loaded catalysts were dispersed into 1 mL of Nafion and ethanol (volume ratio of 1:49) mixture solution, and homogeneous catalyst ink was obtained by ultrasound for 30 min. The thin films of catalysts supported on glassy carbon electrode (GCE) were prepared by using rotational drying method. The GCE were polished to a mirror-finish prior to each experiment and served as substrates for the catalysts. For this method, the GCE is attached to the shaft of the rotator with the GCE surface facing up, and 5 uL of catalyst ink was then pipetted onto the GCE surface. The catalyst ink was dried with a rotation rate of 700 rpm for at least 20 minutes (under ambient conditions) to deposit high-quality catalyst thin film on GCE[48]. The actual Pt loading for all the catalysts was kept at 6.5 μg cm⁻², which was further determined by ICP-OES measurements. All

the loading mass were normalized over the geometric electrode area of 0.196 cm$^2$.

All electrochemical tests for oxygen reduction reaction (ORR) were performed on a CS310H electrochemical workstation, using a three-electrode electrochemical setup with a rotating disk electrode (RDE) system. A glassy carbon RDE (5 mm inner diameter, 0.196 cm$^2$) was used as working electrode, a graphite rod as counter electrode and an Ag/AgCl electrode of the saturated KCl solution and a matching salt bridge (Luggin capillary) as reference electrode. The potentials involved in the experimental procedure are relative to the reversible hydrogen electrode (RHE), and the potential measured by the Ag/AgCl electrode can be converted into the RHE potential by the following equation:

$$E(vs.RHE) = E(vs.Ag/AgCl) + 0.197 + 0.059\,pH \qquad (2)$$

Prior to electrochemical measurements, the cyclic voltammetry (CV) was performed at a potential sweep rate of 100 mV s$^{-1}$ from 0.03 to 1.1 V (vs. RHE) until stable voltammograms were obtained in N$_2$-saturated 0.1 M HClO$_4$ electrolyte. The CV characterization of the catalysts was usually performed in the potential range of 0.03–1.1 V at a scan rate of 50 mV s$^{-1}$ in an N$_2$-saturated 0.1 M HClO$_4$ solution. Linear scanning voltammetry (LSV) was carried out from 0.2 to 1.2 V in an O$_2$-saturated 0.1 M HClO$_4$ electrolyte with a scan rate of 10 mV s$^{-1}$ and various rotation rates with iR-compensation. The accelerated durability testing (ADT) was performed in an O$_2$-saturated 0.1 M HClO$_4$ solution at room temperature. After ADT, we wipe samples from the surface of the working electrode with cotton soaked in ethanol and collect them in a glass vial. The catalysts were then re-dispersed in ethanol by ultrasonication and collected finally by centrifugation.

## Theoretical calculations

Energy calculations in this work were performed using the Vienna ab initio simulation package (VASP)[49,50] based on the density functional theory (DFT). The generalized gradient approximation (GGA-PW91) was chosen as the exchange-correlation functional[51]. The kinetic energy cutoff was set at 450 eV. The Brillouin zone sampling was treated using the Monkhorst-pack grid[52], and a $4 \times 4 \times 1$ Monkhorst-pack K-point mesh was used during the whole calculation. The optimization thresholds were $10^{-5}$ eV and 0.01 eV/Å for electronic and ionic relaxations, respectively.

## Models

The platinum crystal was constructed from the corresponded JCPDS card (01-001-1311), the face center cubic with space group of Fm-3m (No. 225).), with the DFT optimized lattice constant (a = b = c = 3.924 Å, $\alpha = \beta = \gamma = 90°$) and bulky energy of −24.40 eV. Moreover, the extended and/or shrunken Pt lattice was built from 93% to 112% of the optimized lattice, followed by a DFT structure optimization. Then various corresponded slabs of Pt (111) were consisted by 4 atomic layers with a $3 \times 3$ super cell, each slab was separated by a 15 Å vacuum space. After slab optimization ($E_{slab}$), oxygen atom (1.89 eV, $E^{ads}$) was placed at the hollow site of face centered cubic-packed (fcc) on the surface of Pt (111) for further DFT optimization ($E_{slab}^{ads}$). the adsorption energies of O on Pt (111), with various of lattice parameters, were calculated by the following equation:

$$E_{ads} = E_{slab}^{ads} - E^{ads} - E_{slab} \qquad (3)$$

where, $E_{slab}^{ads}$, $E^{ads}$ and $E_{slab}$ are the energies of calculated adsorbates-slab, adsorbates in gas phase and pure slab, respectively.

## Data availability

All relevant data supporting the key findings of this study are available within the article and its Supplementary Information files or from the corresponding author upon reasonable request. Source data are provided with this paper.

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

## Acknowledgements

This work is financially supported by National Natural Science Foundation of China (52072281, Y.L.); The Major Program of the National Natural Science Foundation of China (22293021, Y.L., B.-L.S.); National Natural Science Foundation of China (52103285, Z.Y.H.); The National innovation and entrepreneurship training program for college students (No. S202210497011, S.L, Z.Q and Y.L.). Y.L. gratefully acknowledges Youth Innovation Research Fund project of State Key Laboratory of Advanced Technology for Materials Synthesis and Processing, Wuhan University of Technology.

## Author contributions

Y.L. conceived the idea and designed the experiment. H.J., Z.W.X., Z.W.Y., and Z.Y.H. performed material characterization, data analysis. Z.D., P.W., S.H.F., S.C.L. and Z.D.Q. helped to characterize the samples. S.H.D. and J.F.L. helped to analyze the data. Z.W. provided DFT theoretical calculations. Y.L. wrote the original drafts, reviews and edits. L.Z., L.Q.M., B.-L.S. and D.Y.Z. supervised the project. All authors read the manuscript, discussed it, and approved its content.

## Competing interests

The authors declare no competing interests.
