## [Peer review file · Nature Communications]

REVIEWER COMMENTS

Reviewer #1 (Remarks to the Author):

The authors show the synthesis of mesoporous Pt@Pt-skin Pt₃Ni core-shell framework nanowires (CSFWs). The structural evolution of the materials from Pt nanowire to Pt@Pt-Ni nanowire intermediate, Pt@Pt-Ni alloy core-shell nanowire, Pt@Pt-skin Pt₃Ni CSFWs are thoroughly studied and presented. Materials characterization including electron microscopy, X-ray diffraction, and X-ray photoelectron spectroscopy was carried out to identify the morphological features, crystal structure, and electronic/valence state of the as-prepared samples. Detailed HAADF-STEM was carried out on the Pt@Pt-skin Pt₃Ni CSFWs to investigate the spatial distribution of Pt and Ni elements. Rotating disk electrode (RDE) half-cell study was performed to evaluate the catalytic materials' activity and durability for oxygen reduction reaction (ORR) under acidic conditions. The Pt@Pt-skin Pt₃Ni CSFWs showed very impressive mass activity as well as durability. Although the manuscript presents very compelling results with a well-organized presentation, it does not meet the high standards to be published in a prestigious journal like Nature Communications in its present form. Two major issues need to be taken care of. First, the authors repeatedly claim the Pt-skin on Pt₃Ni structure but failed to provide direct evidence. In the reviewer's opinion, the HAADF-STEM in Figure 3h is problematic with the arguments provided in the point-to-point comments. Second, although showing very impressive ORR performance and durability, the reproducibility of the RDE testing results is not provided. In addition, the presented data sets are not complete, and certain results for certain samples were missing. Therefore, the reviewer suggests the manuscript be rejected and resubmitted after a major revision addressing the two major problems mentioned above and the point-to-point comments provided below.

Point-to-point comments/questions:

1. Line 128: The authors stated "After annealing in argon/hydrogen mixture at 300 °C, the Pt diffraction peaks showed a slight negative shift (about 0.4°), further confirming the successfully generating compressive lattice strain in the Pt-skin shells." From the reviewer's perspective, the peak positions of Pt@Pt₃Ni CSFWs and Pt@Pt-skin Pt₃Ni CSFWs are identical. If the authors are referring to a 0.4° shift of the (111) diffraction, the higher angle diffraction peaks should show more shifting, which is not the case. Even if there is a shift in the peaks as claimed by the authors, the authors need to explain how the slightly shifted diffraction peak can be correlated to the compressive lattice strain experienced by the Pt-skin. The reviewer does not see how they are related.

2. The XRD peaks for Pt-nanowire are significantly sharper compared with other samples. Some explanation is required. The authors should also calculate the average crystalline domain size and compared it with the diameter obtained from the TEM micrographs.

3. The authors failed to describe how the d-band center was obtained from the data presented in Figure 2f. The interpretation of the d-band center is also problematic because the energy of d-band center is referenced to the Fermi level, i.e., a negative number. If the 0 binding energy in Figure 2f is the Fermi level, then the Pt nanowires actually have the lowest d-band center energy of -2.84 eV while the Pt@Pt-skin Pt₃Ni CSFWs have the highest d-band center energy. The “valence band XPS” in the figure caption should be corrected to “valence band UPS”? The authors can refer to the following reference for more details.

Ando, F.; Gunji, T.; Tanabe, T.; Fukano, I.; Abruña, H. D.; Wu, J.; Ohsaka, T.; Matsumoto, F., Enhancement of the Oxygen Reduction Reaction Activity of Pt by Tuning Its D-Band Center Via Transition Metal Oxide Support Interactions. *ACS Catal.* 2021, 11 (15), 9317-9332.

4. Line 158. The author claims the HRTEM images in Fig. 3h confirmed a well-defined Pt-skin formed on top of the ultrathin Pt₃Ni curves framework shells. However, This image is problematic. The inset is a dark field image while Fig. 3h apparently, is a bright field image. In a bright field image, Z-contrast does not exist. So one cannot conclude that the red-doted region is Pt-rich. Furthermore, it is very hard to tell where exactly Fig h is located in the inset. They don't seem to be coming from the same area. So the repetitive claim of Pt skin@ Pt₃-Ni has little supporting evidence. It is suggested that the author tune down their emphasis on the skin structure or provide stronger evidence of the claimed Pt-skin, including those mentioned in the discussion of XRD.

5. Please execute reproducibility studies on the RDE testing. Error bars should be provided in the bar chart (Fig 4d)

6. The performance of Pt/C and Pt nanowires risk underestimation since the poor maintenance of limiting current typically reflects a poorly deposited catalyst film on the RDE electrode. This causes a mass transfer problem and resulted in the early experiencing of diffusion resistance (mixed diffusion/kintecic region) in the catalyst layer. The authors should re-do the measurements for Pt/C and Pt nanowires to provide a reasonable control group.

7. The author should explain how the mass loading of Pt on the RDE is determined since the errors would be amplified twice. A higher loading would give a much better kinetic current in the ORR measurement. Therefore, if the loading was underestimated, the mass activity would be overestimated significantly. The relation between catalyst loading and measured mass activity is typically non-linear. To obtain a fair comparison, the electrode loading should be the same (difference within 10%).

In addition, based on the provided CV curves in Figure 4a, the Pt@Pt-skin Pt₃Ni CSFWs showed a double layer much larger than other samples. Although it is possible that the porous framework may exhibit

more surfaces compared with other types of Pt nanostructures, carbon black should still dominate the double layer region due to the rather low loading of Pt (20wt%). The reviewer highly suspects that there are substantial errors in the determination of the electrode loading. It is recommended, if accessible, that the authors utilized instruments like XRF to directly measure the loading of Pt on the catalyst-coated RDE.

8. The Tafel slopes are rather similar for all the nanowire-based catalysts. How come the performances are so different? In the potential range from 0.8 - 0.96 V, the more active catalysts seem to start from the mass transfer limited region while the less active ones started from the mixed kinetic/diffusion region. How is it possible that a straight line can be obtained for all four catalysts in this potential region in the Tafel plot?

9. How come there the Pt oxide reduction peak is more pronounced during the CO stripping experiment? CO-stripping results should also be provided for other samples.

10. The accelerated durability cycles protocol is composed of 50,000 CV cycles from 0.6 to 1.1 V vs RHE. The reviewer is quite surprised that there is almost no sign of carbon corrosion based on the CV provided before and after the ADT. Some degree of carbon corrosion was observed for the Pt@Pt₃Ni CSFWs/C while severe carbon corrosion occurred on the Pt/C. It would be nice if the authors could explain more on this part. The durability testing results for the Pt nanowires should be provided.

11. What was the process applied to the catalyst after the ADT to recover the recoverable losses? Was it applied to all catalyst samples in the same way?

Reviewer #2 (Remarks to the Author):

In their manuscript entitled "Mesoporous Pt@Pt-skin Pt₃Ni Core-shell Framework Nanowires for High-Efficient Electrocatalysis" Hui Jin et al. report the design and electrocatalytic oxygen reduction activity (ORR) for anisotropic mesoporous Pt@Pt-skin Pt₃Ni core-shell framework nanowires (CSFWs). While the ORR activity and durability of the reported catalyst are very good and the one-pot synthesis appears to be promising, I cannot recommend this manuscript in its actual form for publication in Nature Communications for the following reasons

- Other than the achieved shape of the catalyst, I cannot find new aspects in the research that would justify publication in Nature Communications, the synthesis method (solvothermal) has been reported already for synthesis of shape-controlled NP
- The authors must provide a much more detailed description of the various effects of time/temperature/molar ratios of capping/reducing agents onto the shape of the final catalyst, as for example shown by Yong Yang et al. (J. Phys. Chem. C 2007, 111, 26, 9095-9104), combined with the various achieved structures and possibly a volcano plot of the corresponding ORR activities. Without this essential information, it is very difficult to understand the various factors that lead to the shape and activity of the final catalyst
- While the reported ORR activity appears impressive compared to 20wt% Pt/C, a comparison to other – highly active shape controlled - Pt₃Ni/C catalyst, for example in a tabulated form, is missing, which would allow direct ORR activity comparison with published results. As for example, Pt subnanometer Pt alloy wires with 4.20 A/mg & 5.11 mA/cm² at 0.9 V vs RHE and very high durability over 30000 cycles were already reported in 2017 (Science Advances, <https://doi.org/10.1126/sciadv.1601705>)
- Structural data as for example spatial dimensions lack error information

Specifically, I would like to add the following questions/comments to the authors:

- Lines 77/78: Standard reduction potentials are given for specific conditions (pH) and aqueous solutions of the metals, therefore under the given reaction conditions they vary. Secondly, what exactly triggers the onset of Ni-reduction after the initial Pt-reduction, only the depletion of Pt cations?
- Line 96: To my eye the EDS line profile indicates the presence of Ni in the initially formed Pt wires, while I miss the corresponding EDS mapping images for the wires
- Line 124: The observed XRD splitting can also be interpreted as phase separation or possibly some de-alloying in the synthesis process
- Line 129: This is confusing because the final synthesis product, Pt@Pt-skin Pt₃Ni CSFWs are obtained after the annealing step, to obtain the Pt-skin structure. If a slight negative shift of 0.4° was observed for the final catalyst, why are the data not presented in figure 2 e?
- Lines 155-160: The Pt/Ni distributions are not very clearly visible from the provided HRTEM and EELS mapping images, the red color for Pt is too dark
- Line 164: To prove the role of Ni in the formation of compressive strain of the Pt-skin, the initial Ni content should be varied, and the resulting structures analyzed (c.f., one of my main concerns stated above)
- Line 180: The authors should provide a reference for the assumed reaction for the formation of adsorbed hydroxyl species
- Line 180: The CV figures in figure 4 a are too small to observe any differences in potential onsets

- Lines 222-223: Why would the Pt-skin structure protect the electrocatalyst against further Ni-leaching from the inner region of the framework walls? Did the authors try to leach Ni by electrochemical cycling and try to determine a maximum of cycles after which no further Ni leaching was observed?
- Line 379: What do the 20% refer to, Pt or catalyst load?
- Lines 401-403: What was the preference to carry out the LSV in anodic correction? Was any iR-compensation applied?

Point-by-point responses to the Reviewers' Comments

Reviewer 1#

General Comments

*The authors show the synthesis of mesoporous Pt@Pt-skin Pt₃Ni core-shell framework nanowires (CSFWs). The structural evolution of the materials from Pt nanowire to Pt@Pt-Ni nanowire intermediate, Pt@Pt-Ni alloy core-shell nanowire, Pt@Pt-skin Pt₃Ni CSFWs are thoroughly studied and presented. Materials characterization including electron microscopy, X-ray diffraction, and X-ray photoelectron spectroscopy was carried out to identify the morphological features, crystal structure, and electronic/valence state of the as-prepared samples. Detailed HAADF-STEM was carried out on the Pt@Pt-skin Pt₃Ni CSFWs to investigate the spatial distribution of Pt and Ni elements. Rotating disk electrode (RDE) half-cell study was performed to evaluate the catalytic materials' activity and durability for oxygen reduction reaction (ORR) under acidic conditions. The Pt@Pt-skin Pt₃Ni CSFWs showed very impressive mass activity as well as durability. Although the manuscript presents very compelling results with a well-organized presentation, it does not meet the high standards to be published in a prestigious journal like Nature Communications in its present form. **Two major issues need to be taken care of.** First, the authors repeatedly claim the Pt-skin on Pt₃Ni structure but failed to provide direct evidence. In the reviewer's opinion, the HAADF-STEM in Figure 3h is problematic with the arguments provided in the point-to-point comments. Second, although showing very impressive ORR performance and durability, the reproducibility of the RDE testing results is not provided. **In addition**, the presented data sets are not complete, and certain results for certain samples were missing. Therefore, the reviewer suggests the manuscript be rejected and resubmitted after a major revision addressing the two major problems mentioned above and the point-to-point comments provided below.*

Response:

We appreciate the reviewer for the comments. The two major issues raised by the reviewer have been well settled in the revised manuscript. **First**, extensive efforts have been devoted to well proving the Pt-skin on Pt₃Ni structure by HAADF-STEM images, EDS mapping & line-scanning profile and CO stripping results (**Please see our detailed responses to Comments 1-4**). **Second**, motivated by the reviewer's kind remind, a complete set of ORR tests (including the reproducibility of the RDE testing, etc.) has been supplemented for a better demonstration of the electrochemical results (**Please see our detailed responses to Comments 5-11**). Given these efforts, we believe the revised manuscript reserves the high quality and can meet the high standards in *Nature Communications*.

Comment 1: *Line 128: The authors stated "After annealing in argon/hydrogen mixture at 300°C, the Pt diffraction peaks showed a slight negative shift (about 0.4°), further confirming the successfully generating compressive lattice strain in the Pt-skin shells." From the reviewer's perspective, the peak positions of Pt@Pt₃Ni CSFWs and Pt @ Pt-skin Pt₃Ni CSFWs are identical. If the authors are referring to a 0.4° shift of the (111) diffraction, the higher angle diffraction peaks should show more shifting, which is not the case. Even if there is a shift in the peaks as claimed by the authors, the authors need to explain how the slightly shifted diffraction peak can be correlated to the compressive lattice strain experienced by the Pt-skin. The reviewer does not see how they are related.*

Response:

We thank the reviewer for raising this comment. We are fully aware that the lattice compression in Pt-

skin shell could not be precisely determined by XRD. In the present work, the compressive strain in Pt-skin shell only affects a few atomic layers on the surface of the Pt@Pt-skin Pt₃Ni CSFWs sample, resulting in its absence of significant peak shift of the XRD pattern when compared to Pt@Pt₃Ni CSFWs (please see Supplementary Figure 16). Therefore, it is really difficult to determine lattice compression in Pt-skin shell only by XRD. However, we have demonstrated the existence of Pt-skin as well as its lattice compression by HAADF-STEM images, Aberration-Corrected HRTEM, EDS mapping & line-scanning profile and CO stripping results (**Please see our detailed responses to Comments 4**).

In the revised manuscript, we removed the inaccurate description “After annealing in argon/hydrogen mixture at 300°C, the Pt diffraction peaks showed a slight negative shift (about 0.4°), further confirming the successfully generating compressive lattice strain in the Pt-skin shells.” And, we provided new TEM data and detailed analysis for the Pt-skin structure (as well as its lattice compression) in Figure 3 and Supplementary Figure 18-19.

Comment 2: The XRD peaks for Pt-nanowire are significantly sharper compared with other samples. Some explanation is required. The authors should also calculate the average crystalline domain size and compared it with the diameter obtained from the TEM micrographs.

Response:

We appreciate the reviewer for the kind suggestion. The bimetallic alloy characteristic/crystallinity nature of the Pt nanowire and other core-shell alloyed samples are confirmed by XRD. As shown in Figure 2e, the diffraction peaks of (111), (200), (220) and (311) are indexed to face centered cubic (*fcc*) structured Pt and PtNi alloy (Pt, no. 04-0802 and Ni, no. 04-0850). When compared to pure Pt nanowire, the XRD diffraction peaks of core-shell PtNi alloy CSFWs (Pt@Pt-skin Pt₃Ni CSFWs and Pt@Pt₃Ni CSFWs, etc.) shift to higher angles with wider width (*in consistence with the previous literature, e.g., Chem. Commun., 2021, 57, 623-626; Front. Energy, 2017, 11, 260-267*), **which could be attributed to the decrease of lattice distance when smaller Ni atoms alloyed with Pt atoms in the lattice and the resulting lower crystallinity.**

Moreover, according to the Scherrer equation, the average *crystalline* domain size of Pt nanowires was estimated to be ~3.5 nm, which is consistent with the diameter obtained from the TEM micrograph in Supplementary Figure 2-4.

In the revised manuscript, we added a detailed explanation for the sharper XRD peak of Pt-nanowire compared with other core-shell alloyed samples on Page 7-8.

Comment 3: The authors failed to describe how the d-band center was obtained from the data presented in Figure 2f. The interpretation of the d-band center is also problematic because the energy of d-band center is referenced to the Fermi level, i.e., a negative number. If the 0 binding energy in Figure 2f is the Fermi level, then the Pt nanowires actually have the lowest d-band center energy of -2.84 eV while the Pt@Pt-skin Pt₃Ni CSFWs have the highest d-band center energy. The “valence band XPS” in the figure caption should be corrected to “valence band UPS”? The authors can refer to the following reference for more details. Ando, F.; Gunji, T.; Tanabe, T.; Fukano, I.; Abruña, H. D.; Wu, J.; Ohsaka, T.; Matsumoto, F., Enhancement

Response:

We greatly appreciate the reviewer for pointing out this issue. The d-band center is calculated according to the following equation:

$$d \text{ band center} = \frac{\int_{-\infty}^{E_f} R(E)E dE}{\int_{-\infty}^{E_f} R(E) dE}$$

where E is the binding energy, R(E) is the UPS intensity after background subtraction, E_f is the Fermi energy level, and the calibration of the UPS revealed that the E_f of all samples was approximately 0.

Since the d-band center in our manuscript was calculated from the calibrated UPS data, it was a positive value, which is consistent with the previous literature (e.g. *Nature Mater.* 2007, 6, 241–247; *J. Am. Chem. Soc.* 2006, 128, 8813–8819). As suggested by the reviewer, we also recalculated the d-band center of Pt by integrating the projected d-band density of state up to the Fermi level, and their values are presented in Supplementary Table 2 (*Note:* During the calculation, all the parameters remain unchanged except that the un-calibrated UPS data, and the interval of integration is negative infinity to Fermi energy level.). In particular, the position of the d-band center of Pt@Pt-skin Pt₃Ni CSFWs (-2.74 eV) exhibited ~0.32 eV downshift as compared to Pt nanowires (-2.42 eV), which can significantly improve the ORR activity by reducing the adsorption energies of oxygenated species on the Pt-skin surface (*Nørskov et al predicted that a better ORR electrocatalyst should bind oxygen weaker than Pt by about 0.2 eV lower than that of Pt. Please refer to *Surf. Sci.* 1995, 343, 211–220 and *J. Mol. Catal. A: Chem.* 1997, 115, 421–429.*)

In the revised manuscript, we added the detailed description of d-band center calculation in Methods on Page 21, and we corrected the “valence band XPS” to “valence band UPS” in the Figure 2 caption. In addition, we have well cited the paper “*ACS Catal.* 2021, 11 (15), 9317-9332.” as Reference 45 in the revised manuscript.

Supplementary Table 1. The d-band center of the samples calculated from high-resolution valence band UPS.

Catalysts	Pt NWs	Pt@Pt-Ni CSNWs	Pt@Pt ₃ Ni CSFWs	Pt@Pt-skin Pt ₃ Ni CSFWs
d-band center(eV)	-2.42	-2.49	-2.58	-2.74

Comment 4: Line 158. The author claims the HRTEM images in Fig. 3h confirmed a well-defined Pt-skin formed on top of the ultrathin Pt₃Ni curves framework shells. However, this image is problematic. The inset is a dark field image while Fig. 3h apparently, is a bright field image. In a bright field image, Z-contrast does not exist. So one cannot conclude that the red-dotted region is Pt-rich. Furthermore, it is very hard to tell where exactly Fig h is located in the inset. They don't seem to be coming from the same area. So the repetitive claim of Pt skin@ Pt₃Ni has little supporting evidence. It is suggested that the author tune down their emphasis on the skin structure or provide stronger evidence of the claimed Pt-skin, including those mentioned in the discussion of XRD.

Response:

We thank the reviewer for pointing out our poor demonstration of the Pt-skin structure formed on surface of the ultrathin Pt₃Ni curves framework shells. To unambiguously identify the Pt skin, we employed high angle annular dark-field scanning transmission electron microscopy (HAADF-STEM), in conjunction with energy-dispersive X-ray spectroscopy (EDX), to achieve elemental mapping & line-scanning over a single mesoporous Pt@Pt-skin Pt₃Ni CSFWs.

The high-resolution HAADF-STEM and the corresponding fast Fourier transform (FFT) patterns taken along $[\bar{1}01]$ zone axis (Supplementary Figure 19) show high quality periodic lattice of face-centered cubic (*fcc*) Pt extending across the entire surface of Pt@Pt-skin Pt₃Ni CSFWs, indicating the formation of Pt-skin surface nanostructure. The simulated Pt atomic model along $[\bar{1}01]$ zone axis (with 4.9% compressive lattice contraction) matches very well with the experimental atomic arrangement taken from the selected areas (marked by the rectangles Supplementary Figure 19). **Compared to bulk Pt (111) spacing values (2.27 Å), the intensity profile of Pt-skin shows a much smaller Pt (111) lattice distance (2.16 Å) and represents ~4.9% compressive lattice contraction (Fig. 3c), which is consistent with high-resolution HAADF-STEM result.**

High-resolution HAADF-STEM-EDX elemental mapping and line-scanning analysis further reveal the formation of an ultrathin Pt shell on the surface of the ultrathin Pt₃Ni curves framework shells. **As shown in Figure R1 and Supplementary Figure 18, the EDX mapping and intensity profile of a single mesoporous Pt@Pt-skin Pt₃Ni CSFW reveals that the edges of the Pt₃Ni framework shells are only composed of Pt atoms, without Ni elemental signal.** Moreover, the atomic layer-by-layer EDX line scan on framework shells clearly showed only Pt signal in the 2nd outmost atomic layer (but Pt & Ni mixed signal in the 10th outer atomic layer), further profoundly demonstrated the Pt-skin structure was formed on the surface of ultrathin Pt₃Ni curved framework shells.

Moreover, the successful formation of Pt-skin structure was also identified by electrooxidation of carbon monoxide (CO stripping) and underpotentially deposited hydrogen (H_{UPD}). **On a Pt-skin surface, H_{UPD} exhibits lower surface coverage due to weakened binding, which yields a characteristic ECSA_{CO}:ECSA_{H_{UPD}} ratio of approximately 1.5 (*J. Am. Chem. Soc.* 2015, 137, 15817–15824, *Science*, 2014, 343, 1339-1343).** This has proven to be an easy way to identify formation of the Pt-skin surface structure over Pt₃Ni. **As shown in Figure 4c and Supplementary Table 2, the ratio of ECSA_{CO}:ECSA_{H_{UPD}} of mesoporous Pt@Pt-skin Pt₃Ni CSFWs is calculated to be 1.47, which strongly confirms the formation of Pt-skin surface on mesoporous Pt@Pt-skin Pt₃Ni CSFWs/C.**

In the revised manuscript, we added more convincing HAADF-STEM, EDS mapping and line-scanning data in Figure 3 and Supplementary Figure 18-19. Detailed discussions were highlighted in red on Page 8-9.

Supplementary Figure 19. (a) HAADF-STEM image of an individual mesoporous Pt@Pt-skin Pt₃Ni CSFWs. (b₁-d₁) The corresponding enlarged view of the rectangular marked area in (a). (b₂-d₂) The corresponding FFT pattern of (b₁), (c₁) and (g₁), respectively.

Figure R1. (a, b, c, d) HAADF-STEM images of an individual mesoporous Pt@Pt-skin Pt₃Ni CSFWs and the corresponding EDX mapping images. (e) EDX intensity profile taken from rectangular marked area in (a). (f) layer-by-layer EDX line scan on outmost framework shells marked in (c).

Comment 5: Please execute reproducibility studies on the RDE testing. Error bars should be provided in the bar chart (Fig 4d).

Response:

We appreciate the reviewer for the kind suggestions. As suggested, all catalysts were evaluated for reproducibility of ORR activity in 0.1 M HClO₄ electrolyte by using the optimized ink formulation. Supplementary Figure 22-24 and Supplementary Table 3 show the reproducibility of the CV curves, ORR polarization curves, specific and mass activity for 4 samples (Pt/C, Pt NWs, mesoporous Pt@Pt₃Ni CSFWs and mesoporous Pt@Pt-skin Pt₃Ni CSFWs), and each sample is measured for 5 independent thin-film electrodes in 0.1 M HClO₄ electrolyte. The average mass activity for Pt/C, Pt NWs, mesoporous Pt@Pt₃Ni CSFWs and mesoporous Pt@Pt-skin Pt₃Ni CSFWs was 0.23 ± 0.012, 1.2 ± 0.045, 2.18 ± 0.062 and 6.69 ± 0.083 A mg⁻¹_{Pt}, respectively. The average specific activity for Pt/C, Pt NWs, mesoporous Pt@Pt₃Ni CSFWs and mesoporous Pt@Pt-skin Pt₃Ni CSFWs was 0.33 ± 0.016, 1.92 ± 0.063, 3.2 ± 0.09 and 8.42 ± 0.126 mA cm⁻², respectively. The above RDE testing clearly confirms that the ORR activity for the same catalyst is reproducible and comparable to each other.

In the revised manuscript, we have added the reproducibility data of the RDE testing in Supplementary Figure 22-24 and Supplementary Table 3, and the corresponding description of reproducibility studies was highlighted in red on Page 12. The error bars corresponding to the standard deviation for the 5 independent measurements were also provided in Figure 4d.

Supplementary Table 3. ORR activity comparison of Pt/C, Pt NWs, mesoporous Pt@Pt₃Ni CSFWs and mesoporous Pt@Pt-skin Pt₃Ni CSFWs based on 5 independent thin-film electrodes in 0.1 M HClO₄ electrolyte.

Catalysts	Pt/C	Pt NWs	Pt@Pt ₃ Ni CSFWs	Pt@Pt-skin Pt ₃ Ni CSFWs
MA ₁ (A mg ⁻¹ _{Pt})	0.227	1.245	2.163	6.773
MA ₂ (A mg ⁻¹ _{Pt})	0.218	1.187	2.118	6.692
MA ₃ (A mg ⁻¹ _{Pt})	0.236	1.223	2.183	6.607
MA ₄ (A mg ⁻¹ _{Pt})	0.242	1.204	2.192	6.653
MA ₅ (A mg ⁻¹ _{Pt})	0.232	1.155	2.242	6.725
SA ₁ (mA cm ⁻²)	0.326	1.983	3.171	8.545
SA ₂ (mA cm ⁻²)	0.314	1.899	3.11	8.423
SA ₃ (mA cm ⁻²)	0.339	1.957	3.198	8.294
SA ₄ (mA cm ⁻²)	0.346	1.926	3.229	8.373
SA ₅ (mA cm ⁻²)	0.333	1.857	3.29	8.463
MA (A mg⁻¹_{Pt})	0.23±0.012	1.2±0.045	2.18±0.062	6.69±0.083
SA (mA cm⁻²)	0.33±0.016	1.92±0.063	3.2±0.09	8.42±0.126

1, 2, 3, 4, 5: The test number of mass activity and specific activity for each catalyst based on 5 independent thin-film electrodes.

Comment 6: *The performance of Pt/C and Pt nanowires risk underestimation since the poor maintenance of limiting current typically reflects a poorly deposited catalyst film on the RDE electrode. This causes a mass transfer problem and resulted in the early experiencing of diffusion resistance (mixed diffusion/kinetic region) in the catalyst layer. The authors should re-do the measurements for Pt/C and Pt nanowires to provide a reasonable control group.*

Response:

We appreciate the reviewer for pointing out this issue. Motivated by the reviewer's suggestions, **we re-do the ORR measurements for both commercial Pt/C (20 weight % Pt on Vulcan XC-72 carbon support, Pt particle size 2 to 5 nm) and Pt nanowires.** For the sake of accurate comparison, all the uniform catalyst thin-films over the whole electrodes are prepared by using exactly the same rotational drying method (please see the experiment section for details). This rotational drying method is a simple and reliable for depositing thin layers of electrocatalysts for excellent performance in RDE methodology. Typically, the well-dispersed Pt/C based ink was dried with a rotation rate of 700 rpm for at least 20 minutes (at room temperature in air) to deposit high-quality Pt/C thin film on glassy carbon disk electrodes. **Based on five independently tested commercial Pt/C electrodes, the average mass activity and specific activity for Pt/C was measured to be 0.23 ± 0.012 A $\text{mg}^{-1}_{\text{Pt}}$ and 0.33 ± 0.016 mA cm^{-2} , respectively, which is well consistent with the values reported by previous work (*Science*, 2016, 354, 1410-1414; *Science*, 2019, 366, 850-856. etc.)**

In the revised manuscript, we have added the 5 retest ORR data of *Pt/C and Pt nanowires* in Supplementary Figure 22-24 and Supplementary Table 3, and the corresponding description of rotational drying method was highlighted in red in the experiment section on Page 22.

Comment 7: *The author should explain how the mass loading of Pt on the RDE is determined since the errors would be amplified twice. A higher loading would give a much better kinetic current in the ORR measurement. Therefore, if the loading was underestimated, the mass activity would be overestimated significantly. The relation between catalyst loading and measured mass activity is typically non-linear. To obtain a fair comparison, the electrode loading should be the same (difference within 10%).*

In addition, based on the provided CV curves in Figure 4a, the Pt@Pt-skin Pt₃Ni CSFWs showed a double layer much larger than other samples. Although it is possible that the porous framework may exhibit more surfaces compared with other types of Pt nanostructures, carbon black should still dominate the double layer region due to the rather low loading of Pt (20wt%). The reviewer highly suspects that there are substantial errors in the determination of the electrode loading. It is recommended, if accessible, that the authors utilized instruments like XRF to directly measure the loading of Pt on the catalyst-coated RDE.

Response:

We appreciate the reviewer for pointing out this issue. In our work, **the mass loading of Pt on the RDE was determined by the inductively coupled plasma-optical emission spectrometer (ICP-OES) measurement, which is widely recognized as an effective measurement to quantify the metal loading in the Pt supported electrocatalysts (*Nature Mater.* 2016, 15, 1188–1194; *Science*, 2019, 366, 850-856. *Nature* 2021 598, 76–81. etc.).** Motivated by the reviewer's suggestions, we re-do the ORR measurements for all four samples (Pt/C, Pt NWs, mesoporous Pt@Pt₃Ni CSFWs and mesoporous Pt@Pt-skinPt₃Ni

CSFWs), and the Pt loading for all samples was strictly controlled to be $6.5 \mu\text{g}/\text{cm}^2$ based on ICP-OES measurement. As shown in Figure 4a in the revised manuscript, the new CV curves of the four samples based on the same Pt loading exhibit almost the same CV double layer, which guarantees a fair comparison of the ORR performance for the four samples. In addition, we retested each catalyst for 5 independent thin-film electrodes to ensure the reproducibility of performance data.

In the revised manuscript, we added the new CV curves of the four samples (Pt/C, Pt NWs, Pt@Pt₃Ni CSFWs and Pt@Pt-skin Pt₃Ni CSFWs) based on the same Pt loading in Figure 4a and Supplementary Fig 22.

Comment 8: *The Tafel slopes are rather similar for all the nanowire-based catalysts. How come the performances are so different? In the potential range from 0.8 - 0.96 V, the more active catalysts seem to start from the mass transfer limited region while the less active ones started from the mixed kinetic/diffusion region. How is it possible that a straight line can be obtained for all four catalysts in this potential region in the Tafel plot?*

Response:

We appreciate the reviewer for pointing out this issue. In the previous manuscript, we fitted and extrapolated the Tafel curves to enable the Tafel plots of all catalysts in the same potential region from 0.8 - 0.96 V, so that the Tafel slopes of all samples seem similar. Motivated by the reviewer's suggestion, we retested the ORR performance of all catalysts and recalculated their Tafel slope near half-wave potential region in LSV. As shown in Figure 4b in the revised manuscript, the Tafel plots of specific activity exhibit slopes of 51.9, 63.4, 69.2 and 80.9 mV dec⁻¹ for Pt@Pt-skin Pt₃Ni CSFWs, Pt@Pt₃Ni CSFWs, Pt NWs and Pt/C electrocatalyst, respectively. A considerably smaller Tafel slope achieved in the Pt@Pt-skin Pt₃Ni CSFWs suggests significantly improved kinetic for ORR.

In the revised manuscript, we have added new Tafel plots of the four samples (Pt/C, Pt NWs, Pt@Pt₃Ni CSFWs and Pt@Pt-skin Pt₃Ni CSFWs) in Figure 4b, and the corresponding description of Tafel plot results was highlighted in red on Page 11.

Figure 4b. The Tafel plots of Pt/C, Pt NWs, Pt@Pt₃Ni CSFWs and Pt@Pt-skin Pt₃Ni CSFWs.

Comment 9: How come there the Pt oxide reduction peak is more pronounced during the CO stripping experiment? CO-stripping results should also be provided for other samples.

Response:

We appreciate the reviewer for pointing out this issue. In our previous manuscript, the reduction peak of Pt oxide was found more pronounced in the CO-stripping curve than that in CV curve, which may probably be ascribed to the following 3 factors:

(1) The degree of activation of the catalysts. The activation degree of the Pt@Pt-skin Pt₃Ni CSFWs catalyst prior to CO-stripping testing may probably be a little bit more sufficient than its CV activation in N₂-saturated electrolyte. Therefore, compared with CV test, more Pt active sites exposed on the Pt@Pt-skin Pt₃Ni CSFWs for CO stripping will result in its more pronounced Pt oxide reduction peak.

(2) The degree of N₂ saturation of HClO₄ solution. The CO-stripping and CV are both tested in N₂-saturated 0.1M HClO₄ solution. For CO-stripping experiment, the degree of N₂ saturation of HClO₄ solution may probably be lower than CV, which enables more Pt oxide formation during the positive-direction scanning. Correspondingly, the Pt oxide reduction peak is more pronounced during the CO stripping under negative-direction scanning.

(3) The sealing of the test system. There will be slight changes for sealing of the CO-stripping and CV test system from time to time, which may induce more dissolution of contaminated O₂ in the reaction during the CO-stripping test, finally resulting in the more pronounced Pt oxide reduction peak during the CO stripping under negative-direction scanning.

As suggested by the reviewer, under strictly control of the above 3 test conditions, we retest the CO stripping experiment for all the samples (Pt/C, Pt NWs/C, Pt@Pt₃Ni CSFWs/C and Pt@Pt-skin Pt₃Ni CSFWs/C) and the CO-stripping results (as well as the corresponding ratio of ECSA_{CO}:ECSA_{Hupd}) of were provided in Supplementary Figure 21 and Supplementary Table 2 in the revised manuscript.

Supplementary Fig. 21. CV and CO stripping curves of (a) Pt/C, (b) Pt NWs/C, (c) Pt@Pt₃Ni CSFWs/C, (d) Pt@Pt-skin Pt₃Ni CSFWs/C, respectively.

Supplementary Table 2. Comparison of ECSA_{CO} and ECSA_{HUPD} among the catalysts.

Catalysts	Pt/C	Pt NWs	Pt@Pt ₃ Ni CSFWs	Pt@Pt-skin Pt ₃ Ni CSFWs
ECSA _{co} (m ² g ⁻¹ _{Pt})	70.47	66.85	71.56	116.6
ECSA _{Hupd} (m ² g ⁻¹ _{Pt})	70.04	62.34	68.22	79.45
ECSA _{co} /ECSA _{Hupd} (m ² g ⁻¹ _{Pt})	1	1.07	1.05	1.47

Comment 10: *The accelerated durability cycles protocol is composed of 50,000 CV cycles from 0.6 to 1.1 V vs RHE. The reviewer is quite surprised that there is almost no sign of carbon corrosion based on the CV provided before and after the ADT. Some degree of carbon corrosion was observed for the Pt@Pt₃Ni CSFWs/C while severe carbon corrosion occurred on the Pt/C. It would be nice if the authors could explain more on this part. The durability testing results for the Pt nanowires should be provided.*

Response:

We thank the reviewer very much for this useful comment. Compared to commercial Pt/C and Pt@Pt₃Ni CSFWs/C, we think that the negligible sign of carbon corrosion of Pt@Pt-skin Pt₃Ni CSFWs may originate from the following factors: (1) The unique electronic structure of the Pt-skin surface may result in a lower coverage of oxygenated intermediates because of the weaker oxygen binding strength, which diminishes the probability of Pt dissolution and the carbon corrosion. (2) The anisotropic porous nature of Pt@Pt-skin Pt₃Ni CSFWs configuration may lead to multipoint Pt-skin surface contacts with the carbon support. Such close contacts may not only prevent movement, aggregation and Ostwald ripening process usually observed in other NPs, but also facilitate the binding between the Pt-skin porous CSFWs nanostructures and the carbon support, thereby contributing to the excellent durability and insignificant carbon corrosion. (3) The optimized Pt-skin thickness of at least 5 Pt monolayers (MLs) (Fig. 3f and Supplementary Figure 18 and 19), hinders the loss of subsurface transition metal through the place-exchange mechanism during electrochemical operation, consequently preserving the high intrinsic activity.

Motivated by the reviewer's suggestion, we also evaluated the durability of the Pt nanowires, and the durability testing results for the Pt nanowires were provided in Supplementary Figure 27. After 10,000 CV cycles, the ECSA dropped by 24.7%, the specific activity dropped by 31.3% and together with mass activity dropped by 31.7%.

In the revised manuscript, we added the durability testing results for the Pt nanowires in Supplementary Figure 27, and the corresponding reasonable explanation of comparison of the level of carbon corrosion for Pt/C, Pt@Pt₃Ni CSFWs and Pt@Pt-skin Pt₃Ni CSFWs were highlighted in red on Page 12 and 13.

Supplementary Figure 27. The durability testing results for the Pt nanowires. **(a)** CV evolution before and after 10,000 potential cycles. **(b)** ORR polarization curve evolution before and after 10,000 potential cycles. **(c, d, e)** Mass activity, specific activity and ECSA evolutions, respectively, before and after different potential cycles.

Comment 11: *What was the process applied to the catalyst after the ADT to recover the recoverable losses? Was it applied to all catalyst samples in the same way?*

Response:

We thank the reviewer for raising this comment. In the present work, the accelerated durability testing (ADT) was performed in an O₂-saturated 0.1 M HClO₄ solution at room temperature. After ADT, we wipe samples from the surface of the working electrode with cotton soaked in ethanol and collect them in a glass vial. The catalysts were then re-dispersed in ethanol by ultrasonication and collected finally by centrifugation. Yes, all catalyst samples after the ADT were processed using the same way described above.

Reviewer 2#

General Comments

In their manuscript entitled “Mesoporous Pt@Pt-skin Pt₃Ni Core-shell Framework Nanowires for High-Efficient Electrocatalysis” Hui Jin et al. report the design and electrocatalytic oxygen reduction activity (ORR) for anisotropic mesoporous Pt@Pt-skin Pt₃Ni core-shell framework nanowires (CSFWs). While the ORR activity and durability of the reported catalyst are very good and the one-pot synthesis appears to be promising, I cannot recommend this manuscript in its actual form for publication in Nature Communications for the following reasons.

Response:

We appreciate the reviewer for the comments. In the revised manuscript, we have carefully addressed all the reviewers' comments and accordingly revised our manuscript. We believe the revised manuscript reserves the high quality and can meet the high standards in *Nature Communications*.

Comment 1: *Other than the achieved shape of the catalyst, I cannot find new aspects in the research that would justify publication in Nature Communications, the synthesis method (solvothermal) has been reported already for synthesis of shape-controlled NP.*

Response:

We thank the reviewer for raising this comment. The solvothermal method for synthesis of shape-controlled nanoparticles has aroused intense interest and admittedly widely reported in recent years, **whereas our Mesoporous Core-shell Framework Nanowires (CSFWs) configuration has not yet been reported before.** We designed a unique mesoporous Pt@Pt-skin Pt₃Ni CSFWs with integration of 3D open mesoporous configuration, 1D anisotropic core-shell motif and lattice strained Pt-skin surface. **This rational design of CSFWs nanostructure is unprecedented and endows ultrahigh electrochemical active surface area and activity.** Especially, the CSFWs configuration exhibits superior electrocatalytic stability with a negligible activity decay (less than 3%) after 50,000 cycles, which has proven to be one of the best-known ORR electrocatalysts to date.

Comment 2: *The authors must provide a much more detailed description of the various effects of time/temperature/molar ratios of capping/reducing agents onto the shape of the final catalyst, as for example shown by Yong Yang et al. (J. Phys. Chem. C 2007, 111, 26, 9095-9104), combined with the various achieved structures and possibly a volcano plot of the corresponding ORR activities. Without this essential information, it is very difficult to understand the various factors that lead to the shape and activity of the final catalyst.*

Response:

We greatly appreciate the reviewer for the useful suggestion. As suggested by the reviewer, we cited the “Yong Yang et al., *J. Phys. Chem. C* 2007, 111, 26, 9095-9104” paper as an important reference (ref.34) in our revised manuscript. Motivated by the reviewer's suggestion, various influence factors (such as time/temperature/molar ratios of capping/reducing agents) onto the morphology of the Pt-Ni nanowires catalysts were studied systematically in our revised manuscript.

As shown in Supplementary Figure 11, the time-dependent morphology changes results revealed that the formation of the well-defined Pt@Pt-Ni CSNWs experienced the initial formation of ultrathin Pt nanowires (0.5 h), the deposition of Ni-rich phase onto the performed Pt nanowires (2 h-12 h), and finally complete reduction of Pt/Ni precursors to form nanogourd-string-like Pt@Pt-Ni alloy CSNWs (24 h).

As shown in Supplementary Figure 12, the temperature-dependent morphology changes results revealed that the length of the nanogourd-string-like Pt@Pt-Ni alloy CSNWs increased from tens of nanometers (160°C) to hundreds of nanometers (180°C), and finally grow to a few microns (200°C). However, when reaction temperature further increased to 220°C, an uneven nanogourd-string-like Pt@Pt-Ni alloy CSNWs and nanoparticles mixture will be formed.

The use of CTAB also plays a critical role in determining the morphology of the Pt-Ni nanowires. As shown in Supplementary Figure 13, without the addition of CTAB, only irregular polyhedral nanocrystals (NCs) with an average size of 30 ± 5 nm were produced. After the addition of 10 mg CTAB, some nanogourd-string-like Pt@Pt-Ni CSNWs mixed with NCs were observed. The uniform nanogourd-string-like Pt@Pt-Ni CSNWs (with an average diameters of 24.3 nm) in a very high yield can be obtained with 40 mg CTAB. While further increasing the CTAB amount to 50 mg, the Pt@Pt-Ni CSNWs yield decreased and the morphology becomes irregular. Therefore, it can be concluded that CTAB was the structure-directing agent and a certain amount of CTAB controlled the growth of nanogourd-string-like Pt@Pt-Ni CSNWs.

Glucose as a reducing agent is also critical to the final morphology of Pt@Pt-Ni alloy core-shell nanowires (CSNWs). As shown in Supplementary Figure 14, in the absence of glucose, irregular nanoparticles with agglomeration were obtained; When the amount of glucose increased to 30 mg, Pt-Ni nanowires with uneven morphology started to form; When the amount of glucose further increased to 60 mg, Pt@Pt-Ni alloy CSNWs with uniform morphology were yielded with diameters of around 24 nm. However, an excessive amount of glucose (90 mg) will result in irregular Pt@Pt-Ni alloy nanowires accompanied by serious agglomeration.

We further studied the effect of the Ni/Pt molar ratio on the formation of the structures. As shown in Supplementary Figure 15, only Pt nanowires with a smooth surface were obtained without the addition of Ni(acac)₂. When increasing the Ni/Pt molar ratio to 0.46 by adding 3mg Ni(acac)₂, the Pt nanowires involved into thin Pt@Pt-Ni nanowires. Typical uniform nanogourd-string-like Pt@Pt-Ni CSNWs (with average diameters of 24.3 nm) were obtained with a Ni/Pt molar ratio of about 1.24 [by adding 8 mg Ni(acac)₂]. While further increasing the Ni/Pt molar ratio to 2.14 [by adding 14 mg Ni(acac)₂], the uneven and agglomerated Pt@Pt-Ni CSNWs with a diameter of 50 ± 8 nm were obtained.

Thus, the above catalyst morphology studies confirm that the proper reaction temperature, time, CTAB, glucose and Ni/Pt ratio are responsible for the fine-controlled production of mesoporous Pt@Pt-Ni CSNWs. Moreover, we also carried out the electrocatalytic performance of different mesoporous Pt@Pt-Ni CSNWs samples obtained from different Ni/Pt molar ratio. (All catalyst samples experienced the same acid treatment and heat treatment before the ORR test). The ORR results in Supplementary Fig. 25 showed the volcano-shaped ORR activity relationship, and the

uniform mesoporous Pt@Pt-skin Pt₃Ni CSFWs/C obtained by adding 8 mg Ni(acac)₂ achieves the best ORR performance.

In the revised manuscript, we added TEM images of time/temperature/molar ratios of capping-dependent morphology changes in Supplementary Figure 11-15, and correspondingly provided a much more detailed description of the various effects of time/temperature/molar ratios of capping/reducing agents onto the shape and ORR performance of the final catalyst in Page 6 and Page 7.

Supplementary Figure 11. TEM images of the samples with the same reaction conditions as those of Pt@Pt-Ni alloy CSNWs except the reaction time of (a) 0.5 h; (b) 2 h; (c) 12 h; (d) 24 h.

Supplementary Figure 12. TEM images of the samples with the same reaction conditions as those of Pt@Pt-Ni alloy CSNWs except the reaction temperature of (a) 160°C; (b) 180°C; (c) 200°C; (d) 220°C.

Supplementary Figure 13. TEM images of the samples with the same reaction conditions as those of Pt@Pt-Ni alloy CSNWs except the use of (a) 0 mg CTAB; (b) 10 mg CTAB; (c) 40 mg CTAB; (d) 50 mg CTAB.

Supplementary Figure 14. TEM images of the samples with the same reaction conditions as those of Pt@Pt-Ni alloy CSNWs except the use of (a) 0 mg glucose; (b) 30 mg glucose; (c) 60 mg glucose; (d) 90 mg glucose.

Supplementary Figure 15. TEM images of the samples with the same reaction conditions as those of Pt@Pt-Ni alloy CSNWs except the use of (a) 0 mg Ni (acac)₂; (b) 3 mg Ni (acac)₂; (c) 8 mg Ni (acac)₂; (d) 14 mg Ni (acac)₂.

Supplementary Figure 25. Electrocatalytic performance of different mesoporous Pt@Pt-Ni CSFWs samples obtained from different the Ni/Pt ratio, and all catalyst samples experienced the same acid treatment and heat treatment before the ORR test. The ORR results showed the volcano-shaped activity relationship, and the uniform mesoporous Pt@Pt-skin Pt₃Ni CSFWs/C obtained by adding 8 mg Ni (acac)₂ achieves the best ORR performance.

Comment 3: While the reported ORR activity appears impressive compared to 20wt% Pt/C, a comparison to other – highly active shape controlled - Pt₃Ni/C catalyst, for example in a tabulated form, is missing, which would allow direct ORR activity comparison with published results. As for example, Pt sub-nanometer Pt alloy wires with 4.20 A/mg & 5.11 mA/cm² at 0.9 V vs RHE and very high durability over 30000 cycles were already reported in 2017 (Science Advances, <https://doi.org/10.1126/sciadv.1601705>).

Response:

We gratefully appreciate the reviewer for this valuable comment. Motivated by the reviewer's suggestion, the comparison of the ORR activity of mesoporous Pt@Pt-skin Pt₃Ni CSFWs/C with other Pt₃Ni/C catalysts published in recent years was provided in Supplementary Table 4. Considering comparable Pt loadings, the performance of mesoporous Pt@Pt-skin Pt₃Ni CSFWs/C is among the best reported performance for Pt₃Ni/C catalysts.

In the revised manuscript, we added Supplementary Table 4 on Page 34 in the revised Supporting Information and the corresponding description of the ORR activity comparison results were highlighted in red on Page 12.

Supplementary Table 4. ORR activity comparison of our work with various Pt₃Ni/C catalysts published in recent years.

Catalyst	ECSA (m ² g ⁻¹ _{Pt})	@0.9V vs. RHE		Durability loss (%)			Reference
		Mass activity (A mg ⁻¹ _{Pt})	Specific activity (mA cm ⁻²)	ECSA (m ² g ⁻¹ _{Pt})	@0.9V vs. RHE		
					Mass activity (A mg ⁻¹ _{Pt})	Specific activity (mA cm ⁻²)	
Pt ₃ Ni/C nanoframes	67.2	5.7	NA	NA	NA	NA	42
Mo-Pt ₃ Ni/C ^a	67.5	6.98	10.3	NA	5.5%	6.2%	51
Pt ₃ Ni/C ^a	66.6	1.8	2.7	NA	41%	33%	
Pt ₃ Ni BANWs ^b	60.3	0.546	0.9	52.2%	66.5%	NA	30
4.5nm-Pt ₃ Ni/C ^c	41.67	1.08	2.47	8.2%	53.7%	50%	52
6.0nm-Pt ₃ Ni/C ^c	42.55	1.18	2.61	7.8%	53.4%	49.8%	
6.8nm-Pt ₃ Ni/C ^c	35.86	1.20	2.94	7.7%	51.7%	49.0%	
8.1nm-Pt ₃ Ni/C ^c	34.16	1.79	4.06	1.5%	51.4%	48%	
Pt ₃ Ni/C ^d	36.8	3.21	5.89	NA	24%	NA	53
Au-Pt ₃ Ni/C ^d	35.3	3.08	5.74	NA	4.9%	NA	
Porous Pt ₃ Ni ^e	70.4	0.757	1.006	27.3%	NA	NA	54
Pt ₃ Ni NWS/C-AA	43.8	4.05	NA	NA	NA	NA	55
Pt ₃ Ni/C ^d	27.6	1.761	4.41	10.4%	NA	NA	56
100% t, o-Pt ₃ Ni	62.4	0.53	0.85	NA	NA	NA	57
Pt@Pt-skin Pt₃Ni CSFWs/C^b	79.45	6.69	8.42	1.7%	2.8%	2.9%	This work

NA: not available.

^{a, b, c, d, e}: The durability loss of the catalysts is obtained after 8000, 50000, 4000, 20000 and 6000 potential cycles, respectively.

Comment 4: *Structural data as for example spatial dimensions lack error information.*

Response:

We thank the reviewer very much for the useful suggestion. Motivated by the reviewer's suggestion, the size distribution (with error bar) of initially formed Pt nanowire, Pt@Pt-Ni alloy CSNWs and final mesoporous Pt@Pt-skin Pt₃Ni CSFWs was provided in Supplementary Figure 10 in the revised Supporting Information. As shown in Figure R2 and Supplementary Figure 10, the average diameter of initially formed Pt nanowire, Pt@Pt-Ni alloy CSNWs and final mesoporous Pt@Pt-skin Pt₃Ni CSFWs is 3.3 ± 1 nm, 24.3 ± 3 nm, 20.3 ± 3 nm, respectively.

Correspondingly, we added the size distribution (with error bar) of the samples in Supplementary Figure 10 in the revised Supporting Information and the corresponding description of the structural data was highlighted in red on Page 6.

Figure R2. (a, b, c) TEM images of initially formed Pt nanowire, Pt@Pt-Ni alloy CSNWs and final mesoporous Pt@Pt-skin Pt₃Ni CSFWs, respectively. (d) The corresponding size distribution with error bar.

Comment 5: Specifically, I would like to add the following questions/comments to the authors:

Lines 77/78: Standard reduction potentials are given for specific conditions (pH) and aqueous solutions of the metals, therefore under the given reaction conditions they vary. Secondly, what exactly triggers the onset of Ni-reduction after the initial Pt-reduction, only the depletion of Pt cations?

Response:

We appreciate the reviewer very much for the comments.

On the first question, the reduction potential admittedly varies with reaction conditions. However, for the statement "... Since a more positive reduction potential of the Pt²⁺/Pt (1.18 eV *versus* RHE) relative to Ni²⁺/Ni (-0.257 eV *versus* RHE) ...", we want to emphasize that the reduction potential of Pt ion is positive to Ni ion under the same reaction conditions in our work, inducing a preferential reduction of Pt ions relative to Ni ions.

On the second question, there are two major factors motivating the onset of Ni-reduction after the initial Pt-reduction. (1) The largely depletion of the Pt precursor accelerates the reduction of Ni ions. (2) Previously reduced Pt serves as a catalyst for Ni reduction. *Peng et al.* studied the growth path of octahedral Pt₃Ni nanoparticles by using *in-situ* ambient pressure X-ray photoelectron spectroscopy (AP-XPS) and X-ray absorption spectroscopy (XAS), confirming that previously reduced Pt has a catalytic effect on the reduction of Ni (*Nat. Commun.* 2018, 9, 4485).

For the purpose of further demonstration, we also compared the reduction ability of Pt(acac)₂/Ni(acac)₂ mixture, pure Pt(acac)₂ and pure Ni(acac)₂ under the same reaction conditions, respectively. This method

has also been reported by *Gong, M. et al. before (ACS Catal. 2019, 9, 4488-4494.)*. As shown in Supplementary Figure 1, the reduction of Pt(acac)₂/Ni(acac)₂ mixture and pure Pt(acac)₂ begins at approximately 160°C. However, the pure Ni(acac)₂ cannot be reduced even under high reaction temperature up to 200°C, profoundly demonstrating that Pt²⁺ ions have more positive reduction potential than that of Ni²⁺ ions under the same reaction conditions in our work, and the previously reduced Pt crystal nuclei serves as a catalyst for Ni²⁺ ions reduction.

Correspondingly, we added reduction ability comparison experiment results in Supplementary Figure 1 and we corrected the corresponding description of the reduction ability on Page 4 in the revised manuscript.

Supplementary Figure 1. Photographs of the Pt/Ni precursor solution before and after reaction.

Comment 6: *Line 96: To my eye the EDS line profile indicates the presence of Ni in the initially formed Pt wires, while I miss the corresponding EDS mapping images for the wires.*

Response:

We thank the reviewer for raising this comment. Since Pt and Ni belong to the same main element family, signal interference of Ni is inevitable in EDS line scanning. The Ni signals appearing in the EDS line-scanning profile in Supplementary Figure 4 are actually the background noise. As suggested, in order to profoundly demonstrate the absence of Ni phase in the initially formed Pt nanowires, the corresponding EDS mapping and intensity images were provided (Supplementary Figure 4). As shown in Supplementary

Figure 4, Pt is the most dominant element in the nanowires, while Ni only exists as background noise. In addition, the EDS intensity images illustrate that all Ni element peaks (Ni-K α , Ni-K β , Ni-L β peaks) are background noise. Consequently, we consider that there is no Ni element in the initially formed nanowires.

Correspondingly, we added EDS mapping images for the Pt nanowires in Supplementary Figure 4 and the description of the EDS mapping results were highlighted in red on Page 5. in the revised manuscript.

Supplementary Figure 4. (a-b) EDS elemental mapping of initially formed atomic-jagged Pt nanowire. scale bars in (a) to (b), 3 nm. (c-d) Corresponding EDS intensity images.

Comment 7: Line 124: *The observed XRD splitting can also be interpreted as phase separation or possibly some de-alloying in the synthesis process.*

Response:

We thank the reviewer for raising this comment. Yes, we fully agree with the review that the XRD peaks splitting can be interpreted as phase separation or possibly some de-alloying in the synthesis process. **However, de-alloying is commonly regarded as the preferential dissolution (or removal) of the electrochemically more active component from a bimetallic alloy (*Nature Chem.*2010, 2, 454–460; *Science* 1991, 254, 687–689; *Nature* 2001, 410, 450–453; *Nature* 2006, 430, 707–710).** Whereas, the phase separation is referred to as a multi-component system sometimes separate into several phases with different components and structures when external conditions such as temperature and pressure change. **In the present work, the Pt@Pt-Ni alloy CSNWs were obtained by hydrothermal condition, without any selective etching treatment in a corrosive medium.** Therefore, the observed XRD splitting of Pt@Pt-Ni

CSNWs in Figure 2e is probably caused by phase separation (Pt-rich phase and Ni-rich phase).

Comment 8: *Line 129: This is confusing because the final synthesis product, Pt@Pt-skin Pt₃Ni CSFWs are obtained after the annealing step, to obtain the Pt-skin structure. If a slight negative shift of 0.4° was observed for the final catalyst, why are the data not presented in figure 2 e?*

Response:

We thank the reviewer for raising this comment. As response to the comment 1 raised by Review 1#, the Pt-skin shell could not be precisely determined by XRD. In the present work, the Pt-skin shell only affects a few atomic layers on the surface of the mesoporous Pt@Pt-skin Pt₃Ni CSFWs sample, resulting in its absence of significant peak shift of the XRD pattern when compared to Pt@Pt₃Ni CSFWs (Supplementary Figure 16). Therefore, it is really difficult to determine the Pt-skin structure by XRD results. Whereas, we have demonstrated the existence of Pt-skin by HAADF-STEM images, Aberration-Corrected HRTEM, EDS mapping & line-scanning profile and CO stripping results (Please see our detailed responses to Comments 4 raised by Review 1#)

In the revised manuscript, we removed the inaccurate description “After annealing in argon/hydrogen mixture at 300°C, the Pt diffraction peaks showed a slight negative shift (about 0.4°), further confirming the successfully generating compressive lattice strain in the Pt-skin shells.” And, we added more convincing HAADF-STEM, EDS mapping & line-scanning and CO stripping data to confirm and analyze the Pt-skin structure on the surface of Pt@Pt-skin Pt₃Ni CSFWs in Figure 3 and Supplementary Figure 18-19. Detailed descriptions of Pt-skin structure were highlighted in red on page 8-9.

Comment 9: *Lines 155-160: The Pt/Ni distributions are not very clearly visible from the provided HRTEM and EELS mapping images, the red color for Pt is too dark.*

Response:

We thank the reviewer for raising this comment. As response to comment 4 raised by Review 1#, we have provided new HAADF-STEM, Aberration-Corrected HRTEM, EDS mapping and line-scanning profile in Figure 3 and Supplementary Figure 17-19, which clearly revealed the Pt/Ni distributions within mesoporous Pt@Pt-skin Pt₃Ni CSFWs. The red color for Pt is also highlighted in the new EDS mapping images in Figure 3e and Supplementary Figure 17-19 in the revised manuscript.

Comment 10: *Line 164: To prove the role of Ni in the formation of compressive strain of the Pt-skin, the initial Ni content should be varied, and the resulting structures analyzed (c.f., one of my main concerns stated above).*

Response:

We thank the reviewer for raising this comment. As response to the comment 2, we systematically studied the effect of the Ni/Pt ratio on the formation of the structures (Supplementary Figure 15) and the results show that only Pt nanowires with a smooth surface was obtained without the addition of Ni(acac)₂.

When increasing the Ni/Pt ratio to 0.46 by adding 3mg Ni(acac)₂, the Pt nanowires involved into a thin Pt@Pt-Ni nanowires. Typical uniform nanogourd-string-like Pt@Pt-Ni alloy CSNWs (with an average diameters of 24.3 nm) were obtained with a Ni/Pt ratio of about 1.24. While further increasing the Ni/Pt ratio to 2.14, the uneven and agglomerated Pt@Pt-Ni alloy CSNWs with the diameter of 50 ± 8 nm were obtained. And we also compared the ORR activities of different mesoporous Pt@Pt-Ni alloy CSNWs/C (obtained from different the Ni/Pt ratio, and all samples experienced the same acid treatment and heat treatment before ORR test), which showed the volcano-shaped activity relationship (Supplementary Figure 25). We will further extend our studies to tease out the relationship between Ni/Pt ratio and compressive strain of the Pt-skin (as well as the ORR activity) in our follow-up paper.

Comment 11: *Line 180: The authors should provide a reference for the assumed reaction for the formation of absorbed hydroxyl species.*

Response:

We thank the reviewer very much for the useful suggestion. As suggested, we cited the literature (*Science* 2009, 324, 1302-1305) as an important reference for the formation of the absorbed hydroxyl species ($\text{OH}_{\text{ad}}, 2\text{H}_2\text{O} = \text{OH}_{\text{ad}} + \text{H}_3\text{O}^+ + \text{e}^-$, $E > 0.7$ eV) on page 10 in the revised manuscript.

Comment 12: *Line 180: The CV figures in figure 4a are too small to observe any differences in potential onsets.*

Response:

We thank the reviewer for raising this comment. As response to comment 7 raised by Reviewer#1, we added the new CV curves of the four samples (Pt/C, Pt NWs, Pt@Pt₃Ni CSFWs and Pt@Pt-skin Pt₃Ni CSFWs) based on the same Pt loading in Figure 4a, and we accordingly adjusted Figure 4a to clearly show the difference in potential onsets of CV curves in the revised manuscript.

Comment 13: *Lines 222-223: Why would the Pt-skin structure protect the electrocatalyst against further Ni-leaching from the inner region of the framework walls? Did the authors try to leach Ni by electrochemical cycling and try to determine a maximum of cycles after which no further Ni leaching was observed?*

Response:

We thank the reviewer for raising this comment.

On the first question, The multilayered Pt-skin surfaces are thick enough to prevent the acid electrolyte (or reactive oxide species) from penetrating the inner Pt-Ni layer, which maybe the main reason why Pt-skin structure can protect the electrocatalyst against further Ni-leaching from the inner region of the framework walls. Typically, *Stamenkovic et al.* also studied the effect of multilayered Pt-skin (1-7 atom monolayer) on the performance of Pt-Ni alloy catalysts, confirming that two or three layers of Pt atoms can effectively prevent the leaching of Ni, thus enhancing the catalytic activity and durability (*J. Am. Chem. Soc.* 2011, 133, 14396-14403). In the present work, the Ni content of mesoporous Pt@Pt-skin Pt₃Ni CSFWs/C catalyst before and after potential-scanning cycles was provided in Supplementary Table 5, which showed that the

Ni content remains almost unchanged after 50,000 cycles. Correspondingly, the mesoporous Pt@Pt-skin Pt₃Ni CSFWs/C catalyst exhibits high stability with negligible activity decay after 50,000 cycles (Figure 4e-f). However, without the Pt-skin protection, the Pt@Pt₃Ni CSFWs/C showed a significant Ni content loss, large negative shift (23 eV) in ORR polarization curves, 43.6% loss of mass activity and 38% loss of ECSA (see Supplementary Figure 26) after only 10,000 cycles.

On the second question, yes, we try to leach Ni by electrochemical cycling and try to determine a maximum of cycles after which no further Ni leaching was observed. Typically, the CV curves of Pt@Pt-Ni CSNWs/C with 10K, 15K, and 20K potential-scanning cycles was performed to determine a maximum of cycles after which no further Ni leaching was observed (Supplementary Figure 29 and Supplementary Table 6).

Supplementary Table 5. The Pt and Ni content of Pt@Pt-skin Pt₃Ni CSFWs/C catalyst before and after potential-scanning cycles.

Cycles	Initial	50K
Pt content (%)	76.75	76.93
Ni content (%)	23.25	23.07

Supplementary Table 6. The Pt and Ni content of Pt@Pt-Ni alloy CSNWs/C catalyst before and after potential-scanning cycles.

Cycles	Initial	10K	15K	20K
Pt content (%)	19.92	58.16	72.83	74.08
Ni content (%)	80.08	41.84	27.17	25.92

Comment 14: *Line 379: What do the 20% refer to, Pt or catalyst load?*

Response:

We thank the reviewer for raising this comment. The 20% refers to the Pt loading. The Pt loading on RDE for all the catalyst was kept at 6.5 $\mu\text{g}/\text{cm}^2$ based on ICP-OES measurements.

Comment 15: *Lines 401-403: What was the preference to carry out the LSV in anodic correction? Was any iR-compensation applied?*

Response:

We thank the reviewer for raising this comment. IR compensation was applied throughout the test. The preference to carry out the LSV testing should include the following factors:

(1) **Solution resistance:** Due to the existence of a certain resistance of the solution, a certain voltage drop will be formed between the working electrode and the reference electrode, which will affect the results of the LSV curve test. We mainly adopt the following methods to reduce the liquid junction potential and

control the solution resistance within 3 ohms. First of all, we fit a salt bridge on the reference electrode and let the tip of the salt bridge as close as possible to the working electrode to reduce the ohmic voltage drop (distance should not be less than the outer diameter of the salt bridge, otherwise it will have a shielding effect on the surface of the working electrode). Then we carry out IR compensation to reduce the effect of solution resistance (IR compensation was generally maintained within 90% to avoid self-excitation of the electrochemical workstation).

(2) **Solution environment:** LSV curves were tested in N₂ and O₂ saturated electrolyte, respectively. To eliminate the effect of background current, the final LSV curves are obtained by subtracting the N₂-saturated data from the O₂-saturated data.

(3) **Catalyst preparation:** The catalysts and carbon black should be well mixed by ultrasonic treatment. In the represent work, all the uniform catalyst thin-films over the whole electrodes are prepared by using the same rotational drying method (please see the experiment section for details).

(4) **Other factors:** We utilize Ag/Cl and graphite as reference electrode and counter electrode (anode) respectively and keep the test environment at room temperature as well as maintaining a good sealing of the test system, etc.

REVIEWERS' COMMENTS

Reviewer #1 (Remarks to the Author):

The authors have gone through the reviewer's comments in detail. The two major concerns pointed out by the reviewer have been addressed thoroughly and adequately. The high-resolution HAADF-STEM characterization of the materials indeed provided much stronger evidence for the claimed Pt-skin structure. The RDE testing was also conducted much more rigorously with results of much higher quality. The reviewer thinks the revised manuscript is of high quality and deserves to be published at Nature Communications.

Reviewer #2 (Remarks to the Author):

In the revised manuscript "Mesoporous Pt@Pt-skin Pt₃Ni Core-shell Framework Nanowires for High-Efficient Electrocatalysis" by Hui Jin et al. the authors present a largely revised version of their original manuscript. However, regarding my comment #1 in my first review "Other than the achieved shape of the catalyst, I cannot find new aspects in the research that would justify publication in Nature Communications, the synthesis method (solvothermal) has been reported already for synthesis of shape-controlled NP." I still do not clearly see the novelty that would justify publication in Nature Communications. In their response to my comment, the authors wrote "The solvothermal method for synthesis of shape-controlled nanoparticles has aroused intense interest and admittedly widely reported in recent years, whereas our Mesoporous Core-shell Framework Nanowires (CSFWs) configuration has not yet been reported before." Is the novelty the synthesis method of applying the solvothermal method in combination with a special capping reagent to obtain specifically core-shell nanowires consisting of two different metals or is it the obtained anisotropic mesoporous structure itself?

In 2018, Yiqi Luo et al reported in ACS Applied Materials and Interfaces (ACS Appl. Mater. Interfaces 2018, 10, 40, 34147–34152, <https://doi.org/10.1021/acsami.8b09988>) about "Mesoporous Pd@Ru Core-Shell Nanorods for Hydrogen Evolution Reaction in Alkaline Solution", in which they describe a synthesis route where the mesoporous nanorod structure was obtained starting from Pd nanorods as seeds to synthesize the Pd@Ru core-shell nanorods by a diffusion process during a 12 h reaction period in a Teflon-lined stainless-steel autoclave under 200 °C. While the authors do not explicitly use the term "solvothermal", the procedure they use is, according to the definition "Solvothermal synthesis is defined as a chemical reaction that takes place in a solvent at a temperature higher than the boiling temperature of the solvent in a sealed vessel." (Cited from "Chemical Solution Synthesis for Materials Design and Thin Film Device Applications", 2021, Pages 79-117, Chapter 3 - Deposition of thin films by chemical solution-assisted techniques, Ankit Kashyap et al., <https://doi.org/10.1016/B978-0-12-819718-9.00014-5>) a solvothermal procedure.

In 2012, Michael C. Carpenter et al., reported the solvothermal synthesis of Pt alloy nanoparticles with the focus on Pt-Ni alloys by using DMF as solvent and reductant. They obtained well-faceted predominantly cubic and cuboctahedral nanocrystals of PtNi alloy with high ORR activity. (Michael C. Carpenter et al., J. Am. Chem. Soc. 2012, 134, 8535–8542, dx.doi.org/10.1021/ja300756y) and showed how in variation of reaction parameters, different structures were obtained.

Therefore, in their introduction, the authors should add some more sentences to make more clear and discuss by citing additional literature the difference to published results and the striking novelty in their approach and obtained catalyst structure, respectively.

Response to other comments

#2: The authors have addressed the question very well and now the effect of the various parameters become clear as also their effect on the final catalyst structure and electrochemical ORR activity. About SI figure 20, the K-L plot refers to SI 20a or 19a?

#3: The issue has been addressed

#4: The issue has been addressed

#5: The issue has been addressed

#6: By the additional data provided by the authors it has become clear now that Ni is absent in the initial phase of the reduction process

#7: The issue has been addressed

#8: The issue has been addressed

#9: The issue has been addressed

#10: The way in which a variation of the Ni content effects the formed structures has become clear now.

#11: The issue has been addressed

#12: The issue has been addressed

#13: The issue has been addressed

#14: The electrode loading has been reported now

#15: Additional information has been provided and it became clear now.

Point-by-point responses to the Reviewers' Comments

Reviewer 1#

General Comments

The authors have gone through the reviewer's comments in detail. The two major concerns pointed out by the reviewer have been addressed thoroughly and adequately. The high-resolution HAADF-STEM characterization of the materials indeed provided much stronger evidence for the claimed Pt-skin structure. The RDE testing was also conducted much more rigorously with results of much higher quality. The reviewer thinks the revised manuscript is of high quality and deserves to be published at Nature Communications.

Response:

We appreciate the reviewer for the positive comments and his kind recommendation for publication of our revised manuscript without any further revision in *Nature Communications*.

Reviewer 2#

General Comments

*In the revised manuscript "Mesoporous Pt@Pt-skin Pt₃Ni Core-shell Framework Nanowires for High-Efficient Electrocatalysis" by Hui Jin et al. the authors present a largely revised version of their original manuscript. However, regarding my comment #1 in my first review "Other than the achieved shape of the catalyst, I cannot find new aspects in the research that would justify publication in Nature Communications, the synthesis method (solvothermal) has been reported already for synthesis of shape-controlled NP." I still do not clearly see the novelty that would justify publication in Nature Communications. In their response to my comment, the authors wrote "The solvothermal method for synthesis of shape-controlled nanoparticles has aroused intense interest and admittedly widely reported in recent years, whereas our Mesoporous Core-shell Framework Nanowires (CSFWs) configuration has not yet been reported before." **Is the novelty the synthesis method of applying the solvothermal method in combination with a special capping reagent to obtain specifically core-shell nanowires consisting of two different metals or is it the obtained anisotropic mesoporous structure itself?***

In 2018, Yiqi Luo et al reported in ACS Applied Materials and Interfaces (ACS Appl. Mater. Interfaces 2018, 10, 40, 34147–34152, <https://doi.org/10.1021/acsami.8b09988>) about "Mesoporous Pd@Ru Core-Shell Nanorods for Hydrogen Evolution Reaction in Alkaline Solution", in which they describe a synthesis route where the mesoporous nanorod structure was obtained starting from Pd nanorods as seeds to synthesize the

Pd@Ru core-shell nanorods by a diffusion process during a 12 h reaction period in a Teflon-lined stainless-steel autoclave under 200 °C. While the authors do not explicitly use the term “solvothermal”, the procedure they use is, according to the definition “Solvothermal synthesis is defined as a chemical reaction that takes place in a solvent at a temperature higher than the boiling temperature of the solvent in a sealed vessel.” (Cited from “Chemical Solution Synthesis for Materials Design and Thin Film Device Applications”, 2021, Pages 79-117, Chapter 3 - Deposition of thin films by chemical solution-assisted techniques, Ankit Kashyap et al., <https://doi.org/10.1016/B978-0-12-819718-9.00014-5>) a solvothermal procedure.

In 2012, Michael C. Carpenter et al., reported the solvothermal synthesis of Pt alloy nanoparticles with the focus on Pt-Ni alloys by using DMF as solvent and reductant. They obtained well-faceted predominantly cubic and cuboctahedral nanocrystals of PtNi alloy with high ORR activity. (Michael C. Carpenter et al., J. Am. Chem. Soc. 2012, 134, 8535–8542, [dx.doi.org/10.1021/ja300756y](https://doi.org/10.1021/ja300756y)) and showed how in variation of reaction parameters, different structures were obtained.

Therefore, in their introduction, the authors should add some more sentences to make more clear and discuss by citing additional literature the difference to published results and the striking novelty in their approach and obtained catalyst structure, respectively.

Response:

We appreciate the reviewer for the comments. **The novelty of our work mainly focuses on the well-defined anisotropic mesoporous Pt@Pt-skin Pt₃Ni core-shell framework nanowire structures (CSFWs), which has not yet been reported before.** Such unique CSFWs configuration combines the advantages of 1D ultrathin atomic-jagged Pt nanowire (diameter ~ 3 nm) core and 3D open lattice-strained Pt-skin (~1-1.5 nm) Pt₃Ni framework shell, which exhibits distinctive mass activity (6.69 A/mg_{Pt}) and specific activity (8.42 mA/cm²) toward ORR, nearly 29 and 26 times higher as compared with the state-of-the-art commercial Pt/C catalyst. The catalyst also exhibits high stability with negligible activity decay after 50,000 cycles, which has proven to be one of the best-known oxygen reduction reaction (ORR) electrocatalysts to date.

As mentioned by reviewer, in 2018, *Luo et al.* also reported a 1D mesoporous Pd@Ru core-shell nanorods, which exhibit the competitive hydrogen evolution reaction (HER) catalytic activity and stability. **However, the morphology, structure and bimetallic composition reported by Luo et al are completely different with our Pt@Pt-skin Pt₃Ni CSFWs configuration.** The mesoporous Pd@Ru core-shell nanorod is composed of a porous face centred cubic Pd and an inward expanded distribution of hexagonal close-packed Ru. **In contrast**, our Pt@Pt-skin Pt₃Ni CSFWs

is composed of a 1D ultrathin atomic-jagged Pt nanowires (diameter ~ 3 nm) core and a 3D open lattice-strained Pt-skin (~1-1.5 nm) Pt₃Ni framework shell. Such 3D open CSFWs configuration and compressive-strained Pt-skin surface is confirmed to provide more catalytically active sites and weaken chemisorption of oxygenated species, thus boosting its catalytic activity and stability towards electrocatalysis.

As mentioned by reviewer, in 2018, *Michael K. Carpenter et al.* reported the preparation of well-faceted Pt alloy NCs (including cubic, octahedral Pt-Ni alloy NCs, etc.) with high ORR activity. **However, these solid Pt alloy NCs contain a substantial proportion of noble metals (Pt) in the bulk than at the surface, which limits the noble-metal utilization and their commercial applications.** In contrast, our mesoporous Pt@Pt-skin Pt₃Ni CSFWs configuration can greatly reduce the Pt content with maximizing activity and Pt atomic utilization by exposing both interior and exterior surface.

As suggested, we added some more sentences in the introduction of the revised manuscript to highlight the novelty of our work by citing additional literatures mentioned above on **Page 2 and Page 3.**

Comment 2: The authors have addressed the question very well and now the effect of the various parameters become clear as also their effect on the final catalyst structure and electrochemical ORR activity. About SI figure 20, the K-L plot refers to SI 20a or 19a?

Response:

We thank the reviewer for the positive comment. The K-L plot refers to Supplementary Figure 20a.